# Hierarchical incremental learning deciphers molecular arrangements in multi-component materials

Hanyin Zhang[1,2], Nan Lin [3], Austin M. Evans[4], Tonghui Wang [5], Saied Md Pratik[6], Jean-Luc Bredas [6] & Haoyuan Li [1,2] ✉

Identifying meaningful patterns of atomic and molecular arrangements from molecular simulations is crucial for revealing microscopic mechanisms in materials. Unraveling these patterns is challenging for the multi-component systems frequently encountered in advanced materials, energy and environmental applications. This limits the understanding of the microscopic mechanisms that ultimately govern the performance of devices based on these systems. Here, we propose a hierarchical incremental learning research protocol named HiDiscover to systematically expedite the mechanistic exploration in multi-component materials. As illustrations, we study Li-ion transport and gas adsorption in nanoporous framework materials, as well as molecular packing in organic active layers for photovoltaics. The HiDiscover protocol enables the detailed differentiation and facile extraction of ionic and molecular arrangements, and reveals quantitative microscopic features that are difficult to discern through conventional molecular simulations, thereby informing materials design. Our approach is seen to improve the reliability of mechanistic descriptions for three different processes in three different classes of materials.

A key to developing novel active molecules, materials, or devices in materials, energy, and environmental sciences is the comprehension of the microscopic characteristics and temporal evolution of atomic and molecular systems[1-6]. At this point in time, mechanistic insight (whether focused on static or dynamic features) is largely derived from molecular simulations[7], which determine the evolution of atomic coordinates that are denoted as trajectories. In their raw format, these data do not directly inform any mechanistic process. Therefore, a crucial task for the researcher is to identify meaningful atomic/molecular arrangements, which are required to derive the microscopic mechanisms[8-10]. However, gaining a clear understanding of the features of the often complex atomic/molecular arrangements within

multi-component systems remains difficult, which limits their impact in terms of revealing the microscopic mechanisms[11]. This challenge is becoming increasingly significant as the research interests increasingly involve mixed organic/inorganic composites[12-15] or interfaces[16-19]. The study of these advanced materials is already complicated by slow dynamics and/or complex factors that often hinder full characterization via a single simulation. Consequently, mechanistic insights are frequently inferred from data available to the researcher.

We take two-dimensional covalent organic frameworks (2D COFs) as an example, which have been extensively studied in recent years as solid-state electrolytes in metal batteries for increased safety and lower operating temperatures[20-25]. Understanding the mechanisms

[1]School of Microelectronics, Shanghai University, Shanghai, China. [2]Key Laboratory of Advanced Display and System Applications, Ministry of Education, Shanghai University, Shanghai, China. [3]Department of Statistics and Data Science, Washington University, St. Louis, MO, USA. [4]George and Josephine Butler Polymer Laboratory, Department of Chemistry, University of Florida, Gainesville, FL, USA. [5]Key Laboratory of Automobile Materials, Ministry of Education, and School of Materials Science and Engineering, Jilin University, Changchun, China. [6]Department of Chemistry and Biochemistry, The University of Arizona, Tucson, AZ, USA. ✉e-mail: lihaoyuan@shu.edu.cn

underlying Li-ion transport in these frameworks is critical for their design towards higher ionic conductiviti, yet our knowledge of the microscopic processes involved remains incomplete[24]. The challenge lies not only in the long timescales required to accurately capture the ion motion in the pores of 2D COFs[24], but also in the lack of tools capable of extracting meaningful ionic arrangements from these composite and disordered systems. Currently, the interpretation of the molecular simulation data obtained on such systems heavily relies on a manual examination of trajectories (often incomplete due to the large number of frames that need to be analyzed) and reasoning based on statistical indicators (such as the radial distribution functions) and the researcher's expertise[26–28]. However, this approach generally leads to empirical mechanistic inferences instead of exhaustive and quantitative characteristics. The heavy reliance on the researcher's experience also introduces potential biases, which makes the resulting interpretations susceptible to subjective information.

Methods have been relatively well established to identify atomic arrangements in simple structures like crystals, with earlier approaches exploiting structural descriptors like the common neighbor analysis[29], centrosymmetry parameter[30], common neighbor parameter[31], and others[32–34]; these methods, however, often suffer from reduced accuracy at large deviations from the perfect crystal structures, making them inapplicable to the complex, multi-component materials[35,36]. Recently developed machine-learning (ML) approaches have shown improved accuracy and broader applications to materials[35,37,38] and have facilitated the descriptions of a few complex systems, such as solid-liquid interfaces[39] and amorphous atomic solids[40,41]. A key in such approaches is a dataset of labeled atomic structures built from a series of isolated reference molecular models that represent different types of atomic arrangements; this dataset is used to train the machine-learning model that then predicts the local structure within a target system by similarity. However, due to the inherent challenges in constructing datasets of isolated phases for multi-component materials with diverse interfaces, developing a machine-learning model capable of identifying atomic and molecular arrangements in complex systems remains a difficult proposition[38,39,42]. We note that, although machine-learning approaches have made significant advances in recent years for analyzing molecular simulation data, current methodologies remain inadequate for identifying meaningful atomic/molecular arrangements in complex multi-component materials[43–45].

Herein, we propose an incremental learning[46] approach to allow the use of overlapping datasets in training machine-learning models that identify atomic and molecular arrangements. This makes it feasible to construct reference molecular models for general multi-component materials. In practice, one can use a series of molecular models with similarities to the target system as a reference for machine learning. Our results demonstrate the feasibility of constructing these reference systems for diverse multi-component materials relevant to energy and environmental research applications. The labels can be appropriately defined and learned in an incremental manner by feeding datasets sequentially. We further divide the problem into separate tasks, each dealing with a portion of the whole system, facilitating model training while providing flexibility for focusing on the parts of interest. The predicted labels from different tasks are combined at a later stage, forming a complete descriptor of microscopic atomic and molecular arrangements. The final model has a hierarchical structure with task incremental learning in the outer layer and class incremental learning in the inner layer. As a result, this approach enables us to identify relevant atomic and molecular arrangements in general multi-component materials, addressing a critical challenge that remains intractable using previous methodologies.

In this framework, to systematically accelerate the mechanistic studies of materials, we have developed a research protocol that we named HiDiscover. This protocol allows the analysis of the machine-learned labels of atomic/molecular arrangements directly from the simulations, and reduces the reliance on a researcher's experience in addressing raw simulation data, often in a non-exhaustive way. To illustrate the efficacy of the HiDiscover protocol, we examine three distinct systems: (i) Li-ion transport in a 2D COF[47], (ii) $CO_2$ adsorption in a metal-organic framework (MOF)[48], and (iii) molecular packing in the active layer of an organic solar cell[49]. Our approach brings forth quantitative microscopic insights into these complex systems, which cannot be obtained using conventional approaches. Thus, these representative examples demonstrate the potential of HiDiscover in accelerating mechanistic studies in the realm of materials, energy, and environmental sciences.

## Results
### The incremental learning approach
To discuss the problem in a general framework, we use $\{\mathbf{x_i} \in \mathbf{R}^d; i = 1, 2, ..., n\}$ to denote the space within which our system exists. The distribution of atomic and molecular arrangements typically displays prevailing patterns $\{y_i; i = 1, 2, ..., m\}$, which can be identified through techniques such as clustering[44,50]. However, these results are often difficult to comprehend at the human level, as we tend to describe atomic/molecular arrangements more effectively at an abstract level. For instance, we find it easier to comprehend the contrast between the arrangement of $H_2O$ molecules in liquid and solid states, rather than memorizing the preferable degrees of orientation exhibited by $H_2O$ molecules relative to one another.

With the description outlined above, we can categorize atomic patterns with similar meanings (contexts) to facilitate comprehension. Each context $\mathcal{C}$ is a subset of $\{y_i\}$. For instance, when investigating the molecular arrangements of $H_2O$, we can define a context set $\{\mathcal{C}i; i = 1,2,...,p\}$ consisting of $\mathcal{C}_{liquid}$ and $\mathcal{C}_{solid}$, which encompass patterns in the liquid and solid phases, respectively. These definitions align with the labels commonly employed for classifying atomic arrangements[11,35,39].

The main challenge when dealing with multi-component systems is that their molecular models often include contexts that are difficult to isolate, making the generation of datasets for individual labels challenging. Acknowledging this issue, we introduce $\Omega$ to represent a subset of $\{\mathcal{C}\}$. Then, it becomes feasible to construct molecular models that exhibit similarities to our target system, resulting in a series of context collections denoted by $\{\Omega_i; i = 1, 2,..., q\}$. However, compared to the previous machine-learning approaches that construct non-overlapping datasets with known labels, $\Omega_i$ and $\Omega_j$ will likely overlap and thus have unknown labels, as illustrated in Fig. 1a. As a result, these datasets cannot be directly used to train a machine-learning model. Here, we propose to adopt an incremental learning[46] methodology and feed these datasets sequentially into a model that learns the features in a step-wise manner. In this way, the resulting model discerns dissimilarities between different $\Omega$ contexts. Specifically, this class-incremental learning model identifies the excess context between successive datasets, which can serve as our label; the initial label corresponds to the first dataset and subsequent labels represent the excess contexts between successive datasets: $\{\Omega_1, \Omega_2\text{-}\Omega_1, \Omega_3\text{-}\Omega_2,..., \Omega_q\text{-}\Omega_{q-1}\} = \{A_1, A_2, A_3, ..., A_q\}$. Such a machine-learning model can effectively predict labels in the target system. A further discussion on the relevance of the contexts can be found in Section 4.1. We note that the order of $\Omega$ fed into the model is a factor in determining the labels (see Sections "Molecular dynamics simulations" and "Model training" for details). A more technical description of this approach is provided in Section 1 of the Supplementary Information (SI).

A real-world problem often consists of multiple tasks ($\zeta$), each with its specific context set {A}, {B}, {C},..., which may involve different data formats. For instance, consider a scenario where we want to investigate the molecular arrangements within an $H_2O$ phase with itself and when a chemical compound (impurity) is mixed, which requires

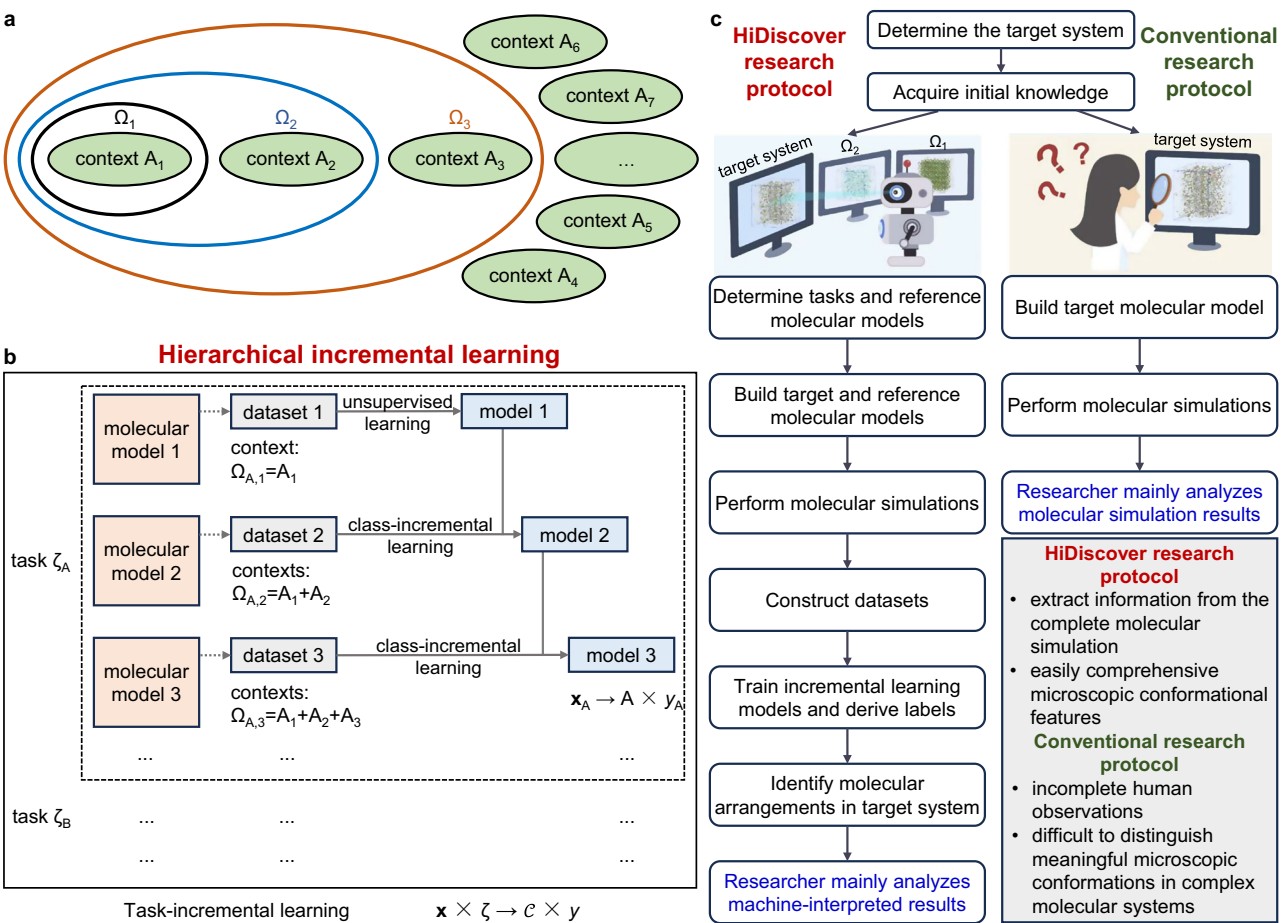

**Fig. 1 | Illustration of the incremental approach. a** Contexts and collection of contexts (highlighted in circles). Context $\mathcal{C}$ represents atomic patterns with similar meanings, and $A_1$, $A_2$,... denote its instances. $\Omega$ represents a subset of the context set $\{\mathcal{C}\}$. **b** The hierarchical incremental learning framework, comprising multiple tasks and each having a class-incremental architecture. $\mathbf{x}$ is a vector representing atomic coordinates, and $y$ denotes a feature. **c** HiDiscover research protocol for simulating multi-component molecular systems, with detailed steps and comparison with a conventional research protocol.

two tasks to be performed. To address such complexity, it is beneficial to incorporate a task-incremental learning model in the outer layer, as illustrated in Fig. 1b. This results in a hierarchical incremental learning framework that is suitable for extracting features of atomic and molecular arrangements in multi-component molecular systems, facilitating a more comprehensive and accurate analysis. To promote its application as a supportive tool in the investigation of microscopic mechanisms in materials, we have developed a research protocol named HiDiscover (Fig. 1c). The detailed steps and comparison with those in a conventional MD study are as follows:

- (i). Identify the material under study and determine the target system for modeling, which is similar to what is done in conventional MD studies. For example, here, we are interested in a metastable $H_2O$-impurity mixture at a given pressure.
- (ii). Acquire initial knowledge about the target system, which can be obtained from literature, prior research experience, or preliminary simulations. In a conventional MD study, the initial knowledge is necessary for the construction of all pertinent molecular models and will inform the researcher of interest points to observe and analyze from the simulation data. In the HiDiscover protocol, this knowledge will also inform the design of tasks and reference systems in the next step.
- (iii). Define tasks and design reference molecular models. This step in the HiDiscover framework differs from conventional protocols. Tasks correspond to specific interest points to which the researcher would pay attention during observation and analysis in

a conventional study, though often implicitly and without documentation (compromising study reproducibility). Here, we explicitly define the tasks for training the machine-learning model. In conventional MD studies, reference molecular systems are also frequently involved, e.g., to use their results as a baseline for comparison; yet, the information gained from this comparison by humans are often limited to intuitive observations and material properties (mobility, diffusivity, etc.) rather than detailed and quantitative atomic/molecular arrangements. When applying the HiDiscover protocol to multi-component materials, the reference molecular models can take the components (such as phases and interfaces) present in the target system while considering the feasibility of model construction. In our example, we may have one task focusing on $H_2O$-$H_2O$ configurations and the other on $H_2O$-impurity configurations. To better extract meaningful labels, the reference molecular models are suggested to have similarities to the target system while demonstrating diversities to allow a fine differentiation of the microscopic configurations. This may require good background knowledge of the studied material and some trial-and-error design, similar to the process of adding molecular systems for comparison in a conventional MD work. We note that, when a HiDiscover protocol is being considered instead of a conventional MD study, a good background knowledge of the studied material (e.g., structural features of its components) is already or will be obtained and a well-defined objective for the researcher to elucidate specific material mechanisms has been or

will be established. As such, the ultimate design of the reference systems is often feasible, as we demonstrate in several distinct examples. Simpler molecular models that incorporate partial components of the studied multi-component material are generally easier to construct than the target system when invoking the HiDiscover protocol (see Section 6 and Table S4 in the SI for further discussions). It is also suggested to incorporate the target system in the reference molecular models to capture potentially missed features. Here, we may build ice, liquid water, and $H_2O$ gas as the reference models in the first task, while having a series of $H_2O$ phases with different impurity concentrations in the second task.

- (iv). Construct the target and reference molecular models, as done in a conventional MD study. Multiple random initial structures need to be considered for inhomogeneous systems.
- (v). Perform molecular simulations for the reference and target molecular models, ideally with long production runs to achieve sufficient coverage of the atomic/molecular arrangements. This step and the considerations are the same as in a conventional MD study.
- (vi). Generate datasets based on the simulation data from the reference and target molecular models.
- (vii). Train each class-incremental model by sequentially feeding its datasets. In this example, we will train two sets of models. In the first task, we may feed datasets of ice, liquid water, and $H_2O$ gas in an incremental learning model. In the second task, we can feed datasets of $H_2O$ with increasing or decreasing impurity concentrations in the other set of models. We note that the assumed order of datasets may need to be tested for a fine differentiation of the contexts (see Notes in Section "Molecular dynamics simulations"). Based on the final order of the datasets, the labels can be derived. Assuming the above-mentioned order of datasets, the three labels in the first task are $H_2O$-$H_2O$ configurations in ice, additional $H_2O$-$H_2O$ configurations in water, and additional $H_2O$-$H_2O$ configurations in $H_2O$ gas.
- (viii). Apply the trained model to the dataset derived from the target system, resulting in a sequence of labels and detailed features of atomic/molecular arrangements. In our example, we apply the two final classification models from the two tasks to the dataset of the target system. We then obtain two sets of labels corresponding to each $H_2O$ molecule in each frame of the simulated trajectory. We can also combine labels from the two tasks to form a complete descriptor of each $H_2O$ molecule for further analysis.
- (ix). Analyze the characteristics of the machine-learned labels, reducing the reliance on researchers directly observing raw MD data.

Steps (vi)−(ix) are all different in the HiDiscover and conventional MD study. We now describe the application of this protocol to three distinct systems, illustrative of current, high-level research on multi-component materials used in energy and environmental applications.

## Li-ion transport in COF-PEO-3

We first use the HiDiscover protocol to investigate Li-ion transport in COF-PEO-3, a representative 2D COF used as a solid-state electrolyte, as shown in Fig. 2[47]. This COF is based on the COF-42 structure with a chemical grafting approach employed to incorporate poly(ethylene oxide) (PEO) chains, which has been shown to enhance Li-ion transport[47,51,52]. A critical step in understanding Li-ion transport in COF-PEO-3 is to know how the cations interact with the various species and their configurations. However, the amorphous nature of the salt-COF composite makes it challenging to identify the configurational features. Here, the HiDiscover protocol enables us to learn the distinctive characteristics of $Li^+$ arrangements when the cations interact

with COF-PEO-3 and the counter-ions $ClO_4^-$. In the latter case, $Li^+$ coordinates with the oxygen atoms of the perchlorate. Regarding COF-PEO-3, it is intuitive to distinguish between ionic arrangements with the main COF framework and with the PEO side chains. Previous work has indicated that $Li^+$ forms stronger interactions with the more electronegative atoms[53,54], specifically: (i) the nitrogen atoms in the COF linkers; (ii) the oxygen atoms at the COF-PEO connections; and (iii) the oxygen atoms in the side chains (Fig. 2). Accordingly, we define four tasks, corresponding to $Li^+$-$ClO_4^-$ ($\zeta_A$), $Li^+$-COF framework linkage ($\zeta_B$), $Li^+$-COF side-chain connection ($\zeta_C$), and $Li^+$-PEO chain ($\zeta_D$).

We then design the reference molecular models for each task and determine the contexts, aiming to ensure configurational diversity, easy human comprehension, and the practical generation of datasets through molecular simulations (see Fig. 2 and Section "Design of the contexts for Li-ion transporting in COF-PEO-3" for details). The chosen reference molecular models comprise crystalline $LiClO_4$, amorphous $LiClO_4$, a $LiClO_4$ cluster in vacuum, $LiClO_4$ in COF-42 (which we recall shares the same framework as COF-PEO-3 but lacks the side chains) with varying concentrations, $LiClO_4$ blended with PEO, and the target system $LiClO_4$ in COF-PEO-3. Table 1 summarizes the definitions of tasks and contexts. We note that the target system is set as the last one in the reference molecular models; thus, it is expected to capture the features missed by the previous reference molecular models. Details on the construction and simulation of these MD models, data processing, and model training can be found in "Methods" and Sections 2–4 of the SI.

The observations derived from the exhaustive machine-learned interpretations offer in-depth insights into the Li-ion configurations in 2D COFs, which would not be available via a conventional analysis of the MD simulations. Specifically, we can now quantitatively describe the ionic configurations in COF-PEO-3, expressed in terms of the labels derived from the reference molecular models we have designed. We first examine the main features of the $Li^+$-$ClO_4^-$ configurations (task $\zeta_A$) in the COF-PEO-3 target system. Figure 3a shows the t-Distributed Stochastic Neighbor Embedding (t-SNE) plot of all $Li^+$-$ClO_4^-$ configurations and the ratios corresponding to different labels. As can be seen, there is negligible (0.2%) crystalline packing ($A_1$), while 1.4% of the total $Li^+$-$ClO_4^-$ configurations correspond to configurations in amorphous $LiClO_4$, which display deformations from the crystal structure (see $A_{2.1}$ in Fig. 3a). The configurations similar to those at the $LiClO_4$ cluster surfaces constitute 7% of the total $Li^+$-$ClO_4^-$ configurations, with oxygen atoms coordinating on one side of the Li-ion, i.e., highly asymmetric ionic configurations (see $A_{3.1}$ in Fig. 3a). Approximately 10% of the $Li^+$-$ClO_4^-$ configurations in COF-PEO-3 resemble those found in mixtures of $LiClO_4$ and PEO, typically with cations and anions separated by large distances (see $A_{4.1}$ in Fig. 3a). Furthermore, 39% of the $Li^+$-$ClO_4^-$ configurations in COF-PEO-3 mirror the patterns observed when $LiClO_4$ is introduced into COF-42, involving 2-3 anions coordinated to the Li-ion (see $A_{5.1}$–$A_{5.3}$ in Fig. 3a). Finally, 42% of $Li^+$-$ClO_4^-$ configurations exhibit new features in context $A_6$ (see $A_{6.1}$–$A_{6.2}$ in Fig. 3a); these configurations demonstrate closer ionic proximity compared to $A_4$ but weaker coordination than in $A_5$, which suggests that introducing the PEO side chains promotes ionic dissociation.

The configurations of $Li^+$-framework linkages (task $\zeta_B$) and $Li^+$-side-chain connections (task $\zeta_C$) share similar trends. As Fig. 3b, c shows, approximately 40% of Li ions are in proximity to the main COF framework ($B_{1.1}$, $B_{1.2}$, $B_{1.3}$, $C_{1.1}$, and $C_{1.2}$), interacting with 1−2 functional groups, typically adopting an in-plane or between-plane configuration. $B_{2.1}$, $B_{2.2}$, $C_{2.1}$, and $C_{2.2}$, which account for ~50% of the configurations, feature a large separation from the main COF framework. The remaining 10% of Li-ion arrangements in COF-PEO-3 demonstrate an even larger separation from the main COF framework ($B_{3.1}$, $B_{3.2}$, $C_{3.1}$, $C_{3.2}$).

Regarding the $Li^+$-PEO configurations (task $\zeta_D$), approximately 24% of them in the target system correspond to context $D_1$; it displays two

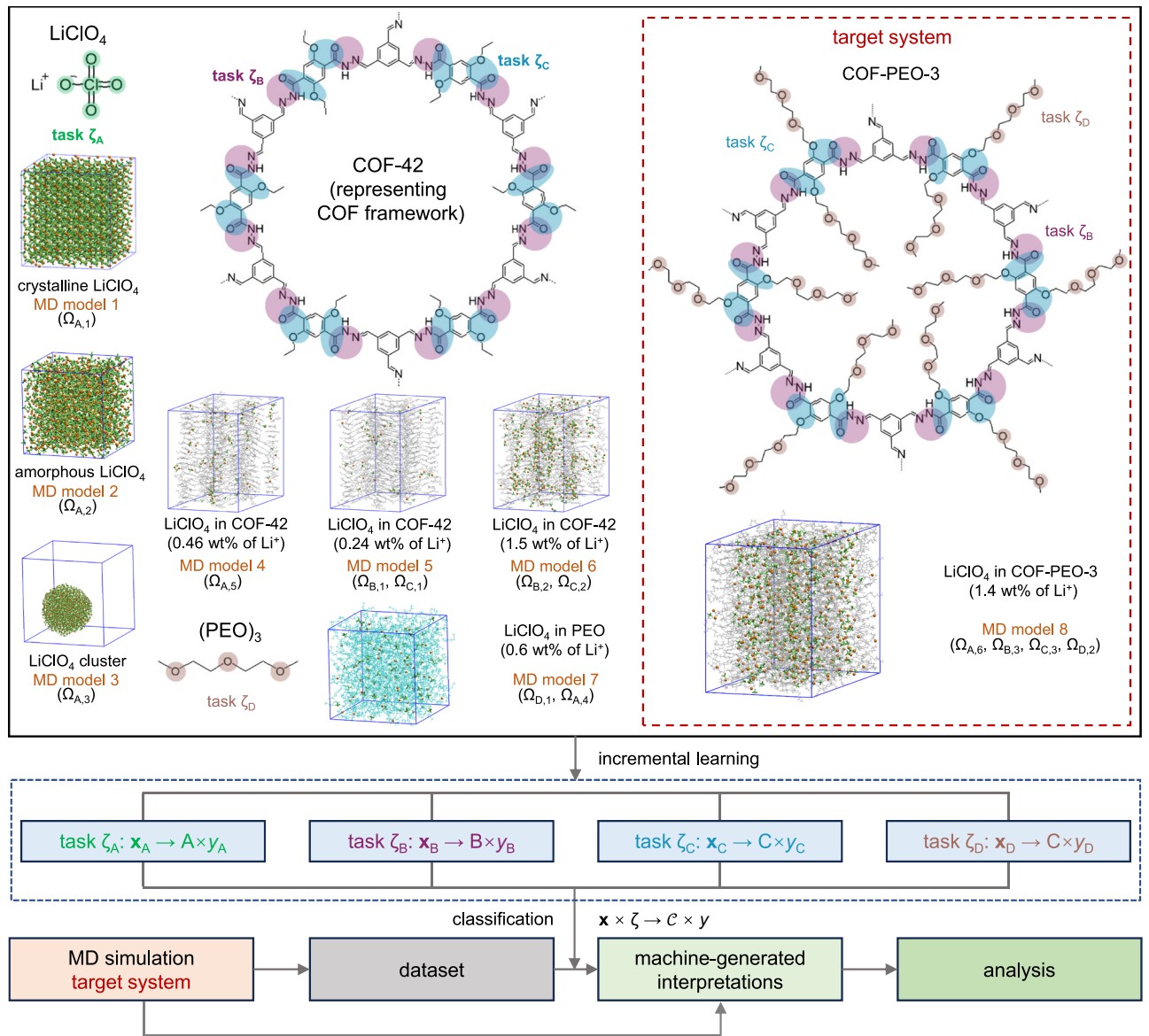

**Fig. 2 | Illustrations of the reference and target molecular models, tasks, and labels in the HiDiscover protocol for studying Li-ion transport in the covalent organic framework COF-PEO-3 containing poly(ethylene oxide) (PEO) chains.** MD models 1–3 correspond to crystalline and amorphous $LiClO_4$ phases and a $LiClO_4$ cluster, respectively. Molecular dynamics (MD) models 4–6 correspond to $LiClO_4$ blended in COF-42 with various weight ratios. Structures with smaller weight ratios capture the arrangements close to the 2D COF framework, while those with higher ratios explore the arrangements in the 2D COF pores. MD model 7 is $LiClO_4$ blended in pure $(PEO)_3$. MD model 8 is the target system. Circles in the chemical structures highlight the regions in the 2D COFs and $(PEO)_3$ that are expected to interact strongly with $Li^+$ (green, purple, blue, and brown circles correspond to tasks $\zeta_A$, $\zeta_B$, $\zeta_C$, and $\zeta_D$, respectively). Further information is provided in Sections 2–3 of the SI. The bottom part of the figure illustrates the process of model training and applying the trained model in data analysis in the HiDiscover protocol.

patterns, as illustrated in Fig. 3d: $D_{1,1}$, with two PEO chains tightly locking the Li-ion; $D_{1,2}$, where one chain displays reduced interactions. The rest of the configurations ($D_{2,1}$, $D_{2,2}$, $D_{2,3}$) exhibit larger $Li^+$-PEO distances, with the closest PEO chains either enveloping the Li-ion or positioned adjacent to it. This underscores the restrictions imposed by the COF framework on the PEO chains due to their covalent bonding, leading to the formation of a loosely bound PEO matrix for the Li ions as compared to the tighter binding found in salt-containing PEO electrolytes.

The combination of the contexts from the four tasks comprehensively depicts the microscopic states of the Li-ions. Overall, the prevailing state observed is $(A_5,B_1,C_1,D_2)$, representing ~21% of all Li-ion arrangements (Fig. 4a); it corresponds to Li ions close to the main COF framework and loosely bound to the PEO side chains. State $(A_6,B_1,C_1,D_2)$, which has slightly weaker $Li^+$-$ClO_4^-$ interactions, accounts

for roughly 12% of total arrangements. Three other states, $(A_6,B_2,C_2,D_1)$, $(A_5,B_2,C_2,D_2)$, and $(A_6,B_2,C_2,D_2)$, each encompassing ~8%–14% of the total arrangements, correspond to the ions away from the main COF framework. Collectively, the five states we just discussed comprise ~65% of all Li-ion arrangements in COF-PEO-3.

These complex features of the Li-ion configurations point to transport mechanisms distinctive from those found in traditional polymer electrolytes. By further analyzing the time sequence of the machine-learned labels, we can get insight into the Li-ion dynamics. We computed the in-state mean square displacement (MSD) of the Li ions (Fig. 4b) and the transition times between various states (Fig. 4c). A detailed analysis is provided in Section 5 of the SI. Overall, our analysis based on the machine-learned labels indicates that the Li-ion motion in COF-PEO-3 at room temperature is impacted by short-range interactions with the PEO side chains, affected indirectly by the 2D COF

framework, and influenced by the Coulombic forces of the surrounding ions. This insight informs the future design of 2D COFs in terms of the following three aspects: (1) side-chain and decoration-group design to control the short-range motions of the Li-ions; (2) framework structure engineering that balances enhancing thermal vibrations that promote hopping among sites and anchoring functional groups that restrict long-range ion migration; and (3) the choice of Li salt (and other ionic groups) to adjust Coulombic interactions for efficient ionic movement along the pores.

We note that the quantitative assessment of the Li-ion configurations and motions discussed above correspond to COF-PEO-3 with 1.38 wt% $Li^+$ (a value close to those used experimentally); the results will change as a function of the $Li^+$ ratio. Notably, a significant ratio of

ions is associated with contexts $A_4$ and $D_1$ at low Li-ion weight ratios (reaching probabilities of ~80% and ~60% at 0.54 wt%, respectively), as shown in Fig. 4d; these correspond to Li-ions well blended in the PEO matrix. We note that this quantitative insight has been overlooked in previous simulations of Li-ion transport in COFs[24]. Our machine-learning interpretations of the MD simulations point out that Li-ions prefer to bind at first to the PEO side chains when adding the Li salt to COF-PEO-3; adding more Li salt leads to additional Li-ions that loosely bind to PEO chains with non-uniform ionic configurations, a process that can even reduce the number of Li-ions uniformly blended and tightly bound to PEO likely due to modifications to the PEO side-chain configurations. Thus, these results underline that it would be worth exploring experimentally the interactions of Li-ions with the various components of 2D COFs as their concentration increases. Furthermore, increasing the Li-ion concentrations leads to a reduction of the diffusivity, which indicates that there is a concentration with optimal ionic conductivity (Fig. S124). Our work highlights that salt concentration is an intrinsic factor in Li-ion transport in 2D COFs, which needs to be considered in materials design instead of tuned empirically as an extrinsic factor, as done so far. In particular, the optimization of Li-ion transport at different salt concentrations should focus on their corresponding Li-ion-COF interactions.

### Table 1 | The definitions of tasks and contexts for studying Li-ion transport in COF-PEO-3

| Task | Context/label | Meaning |
|---|---|---|
| $\zeta_A$ | $A_1$ | configurations in crystalline $LiClO_4$ |
| | $A_2$ | new configurations in amorphous $LiClO_4$ |
| | $A_3$ | new configurations on cluster surfaces |
| | $A_4$ | new configurations when blended in PEO |
| | $A_5$ | interfacial configurations in COF-42 |
| | $A_6$ | new interfacial configurations in COF-PEO-3 |
| $\zeta_B$ | $B_1$ | configurations near the COF framework |
| | $B_2$ | configurations away from the COF framework |
| | $B_3$ | new configurations in COF-PEO-3 |
| $\zeta_C$ | $C_1$ | configurations near the COF framework |
| | $C_2$ | configurations away from the COF framework |
| | $C_3$ | new configurations in COF-PEO-3 |
| $\zeta_D$ | $D_1$ | configurations when $LiClO_4$ is blended in PEO |
| | $D_2$ | new configurations in COF-PEO-3 |

### $CO_2$ adsorption in MOF-5

Gas adsorption in metal-organic frameworks has been extensively studied, as it is closely related to separation, catalysis, sensing, and environmental applications[48,55-58]. While molecular simulations have been useful in predicting diffusivities and adsorption capacities, the microscopic configurational features obtained from these studies, which are needed to fully unravel the adsorption and transport mechanisms, remain limited[55,59-64]. Here, we demonstrate that using the HiDiscover protocol allows us to reveal in detail and quantitatively the arrangement features of $CO_2$ in the representative MOF-5. Specifically, we focus on understanding the features related to $CO_2$-$CO_2$ and $CO_2$-MOF-5 configurations, which will provide insight into

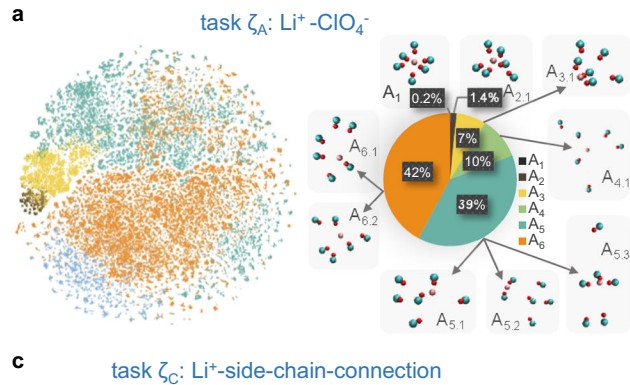

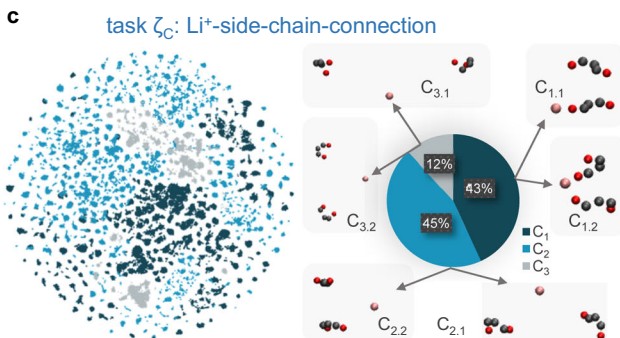

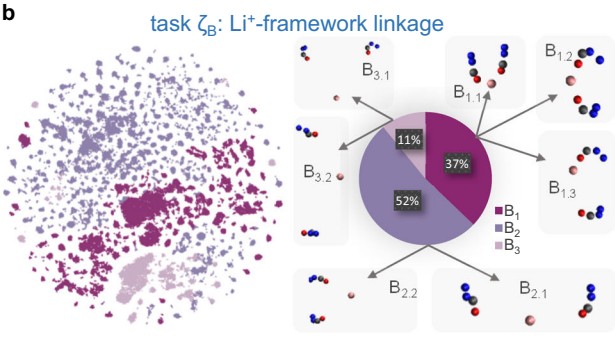

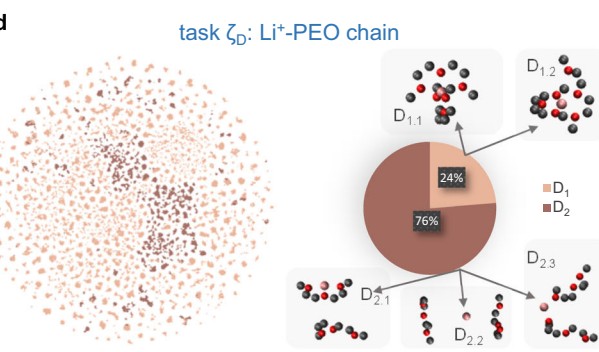

**Fig. 3 | Predicted labels for the Li-ion configurations in COF-PEO-3 in tasks $\zeta_A$-$\zeta_D$ (see Table 1).** The left panels in **a**−**d** provide a t-SNE visualization of the ionic configurations in tasks $\zeta_A$-$\zeta_D$; labels in each task are differentiated according to color. The right panels in (**a**−**d**) show the ratios of different labels in each task and representative arrangements of Li-ions with various species for different labels ($X_Y$) and sub-labels ($X_{Y,Z}$). Pink, red, cyan, gray, and blue spheres represent Li, O, Cl, C, and N, respectively. Visual illustrations of the distributions of Li-ions in tasks $\zeta_B$ and $\zeta_C$ can be found in Fig. S112.

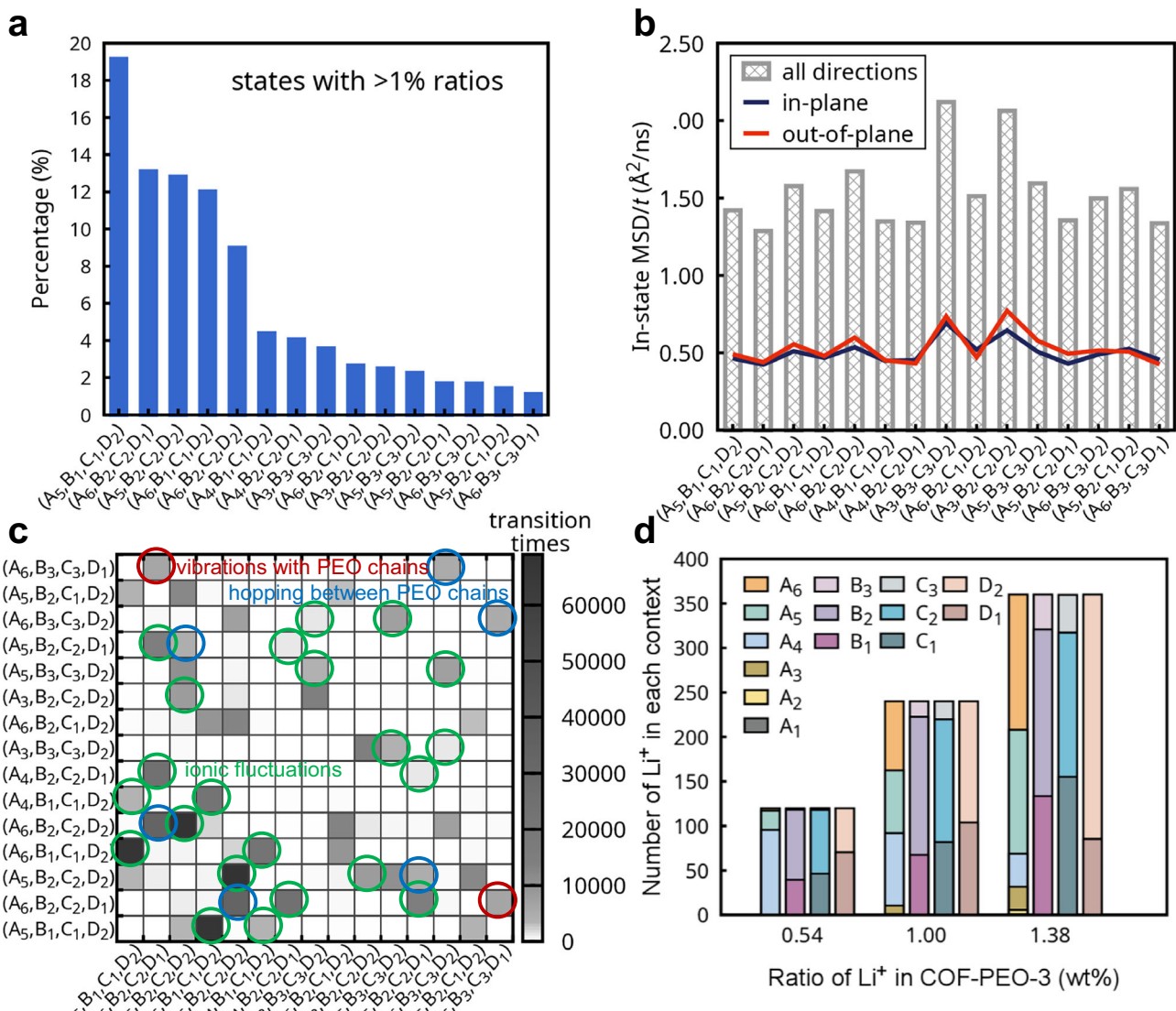

**Fig. 4 | Analysis of Li-ion transport in the covalent organic framework COF-PEO-3 based on machine-learned labels. a** Ratios of different microscopic states (combinations of the contexts from all four tasks) among all states (showing as percentages) of Li-ions in COF-PEO-3 at 1.38 wt%. The total number of states identified in the data is 63. **b** In-state mean square displacements (MSDs) of Li ions per unit time. The gray bars display the MSD value for all three directions in space; red and dark blue lines illustrate the values in the $z$ and in-plane ($x/y$) directions, respectively. **c** Transition times, i.e., number of transitions, (represented by the

color bar) between two states derived from the molecular dynamics (MD) simulations (4 random initial configurations, each with 1-μs simulation time). The area highlighted by circles denotes Li-ion motion that can be interpreted as vibrations with PEO chains, hopping between PEO chains, and ionic fluctuations based on their initial and final states. **d** Number of Li-ions belonging to different contexts at different weight ratios of Li[+] in COF-PEO-3; 0.54 wt%, 1.00 wt%, and 1.38 wt% correspond to a total of 120, 240, and 360 Li[+] in the MD models. Refer to Fig. S123 for further information.

$CO_2$ transport within the MOF. Such information is difficult to visualize by the eye as the gas molecules form dynamic and transient structures from which it is difficult to generate statistically meaningful knowledge. Here, the first task (labeled $\zeta_E$) captures the features related to $CO_2$-$CO_2$ configurations (Fig. 5a), using the $CO_2$ gas at different pressures (1000-1 bar) as reference. The contexts $E_1$–$E_4$ correspond to $CO_2$ packings with densities of ~1–0.003 g/cm³. They act as density indicators and clearly depict the evolution of a denser $CO_2$ packing as the $CO_2$ loading into MOF-5 increases (Fig. 5b). The second task (labeled $\zeta_F$) describes the $CO_2$-MOF configurations: $F_1$–$F_3$ correspond to those that emerge at 0.5, 2, and 10 $CO_2$ mmol/g loadings, respectively.

$F_1$ is found to cover most of the $CO_2$-MOF configurations even at much higher $CO_2$ loadings. In other words, increasing the $CO_2$ loading does not lead to new $CO_2$-MOF configurations. A more detailed analysis shows $F_1$ has four features: $F_{1.1}$ (in-pore corner configurations), $F_{1.2}$

(off-corner configurations, forming pathways connecting adjacent pores), $F_{1.3}$ (in-pore and between-face-center pathways), and $F_{1.4}$ (through-face-center pathways and enveloping $F_{1.3}$), as illustrated in Fig. 5c. We obtain that about half (~46–49%) of the $CO_2$ molecules are in off-corner configurations, suggesting that $CO_2$ prefers to transport around the metal oxide centers both in-pore and to adjacent pores. This result aligns with the human observation that $CO_2$ tends to coordinate with the $Zn_4O_6^+$ cores[64]. Our further analysis reveals that the off-corner configurations can act as a bridge among the other three configurations during $CO_2$ transport (Fig. S125). As the $CO_2$ loading increases, the ratio of molecules in off-corner configurations decreases while that in through-face pathways increases, suggesting that more $CO_2$ is transported to adjacent pores through the pore center (Fig. S126). As such, the HiDiscover protocol enables a detailed differentiation of the molecular arrangements of $CO_2$ in MOF-5, which offers an insight not available from conventional molecular simulation

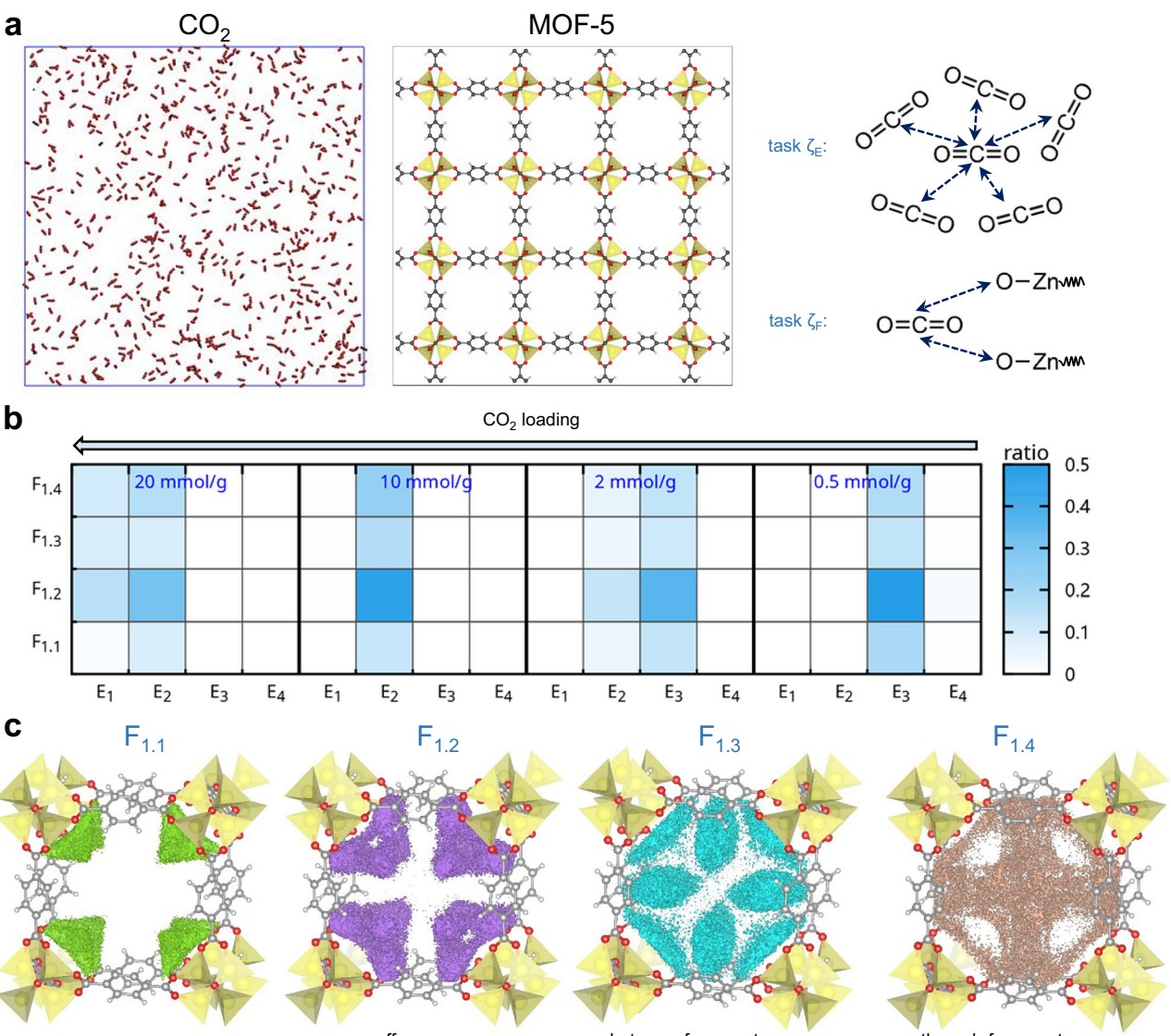

**Fig. 5 | Illustration of the application of the HiDiscover protocol to $CO_2$ adsorption in MOF-5. a** Molecular models of $CO_2$ (10 bar) and MOF-5, and illustrations of tasks $\zeta_E$ ($CO_2$-$CO_2$ configurations) and $\zeta_F$ ($CO_2$-MOF configurations). **b** Fractional ratios (represented by the color bar) of different contexts (among all contexts) for 0.5, 2, 10, and 20 mmol/g $CO_2$ loading in MOF-5. Contexts $E_1$-$E_4$ correspond to $CO_2$-$CO_2$ packing with densities of 1.05, 0.23, 0.018, and 0.003 g/cm³, respectively (we note that $F_2$ and $F_3$ have negligible contributions and are not shown; see Fig. S127 for further information). **c** Illustrations of the spatial distributions of sub-contexts $F_{1.1}$, $F_{1.2}$, $F_{1.3}$, $F_{1.4}$, representing coner, off-corner, between-face-center, and through-face-center configurations, respectively.

protocols and can inform the molecular design of MOFs for gas adsorption, transport, and separation.

## Molecular packing in the active layer of the PM6:Y6 organic solar cell

Finally, we illustrate the use of the HiDiscover protocol to investigate the molecular packing of Y6, a state-of-the-art non-fullerene small-molecule acceptor that has been extensively utilized in binary or ternary active layers in organic photovoltaics[65]. However, the microscopic states of Y6 during device operation remain poorly understood, complicated further by its complex molecular structure that makes the elucidation of the electron transport pathways challenging[66,67]. In the active layers of the bulk heterojunction organic solar cells, which correspond to blends of electron-donor and electron-acceptor components, various domains with different Y6 ratios exist. Hole-electron pairs are often generated at the donor-acceptor interfaces or in mixed phases; these holes and electrons need to transport to their respective collecting electrodes efficiently for the solar cell to achieve high performance. Currently, the Y6 molecular configurations and the electron-transport pathways they form in different regions of the solar-cell active layers remain poorly understood. Here, we examine blends of PM6 (a widely used donor polymer) with Y6 at varying donor-acceptor ratios (Fig. 6a) to probe the electron transport pathways from a donor-rich domain to a mixed domain and then to an acceptor domain, corresponding to the direction of electron transport to the electrode. We note that relying on a conventional molecular simulation protocol is expected to miss the quantitative connections between the various regions in the blend. Here, the HiDiscover protocol enables us to establish quantitatively these connections among domains across space, which allows for a better characterization of the electron-transport pathways over long distances and advances our understanding of global morphological features in organic photovoltaic active layers. Our analysis focuses on the short-range Y6-Y6 contacts (task $\zeta_G$), which is an indicator for the electron transport pathways (as well as for potential charge generation within acceptor domains)[49].

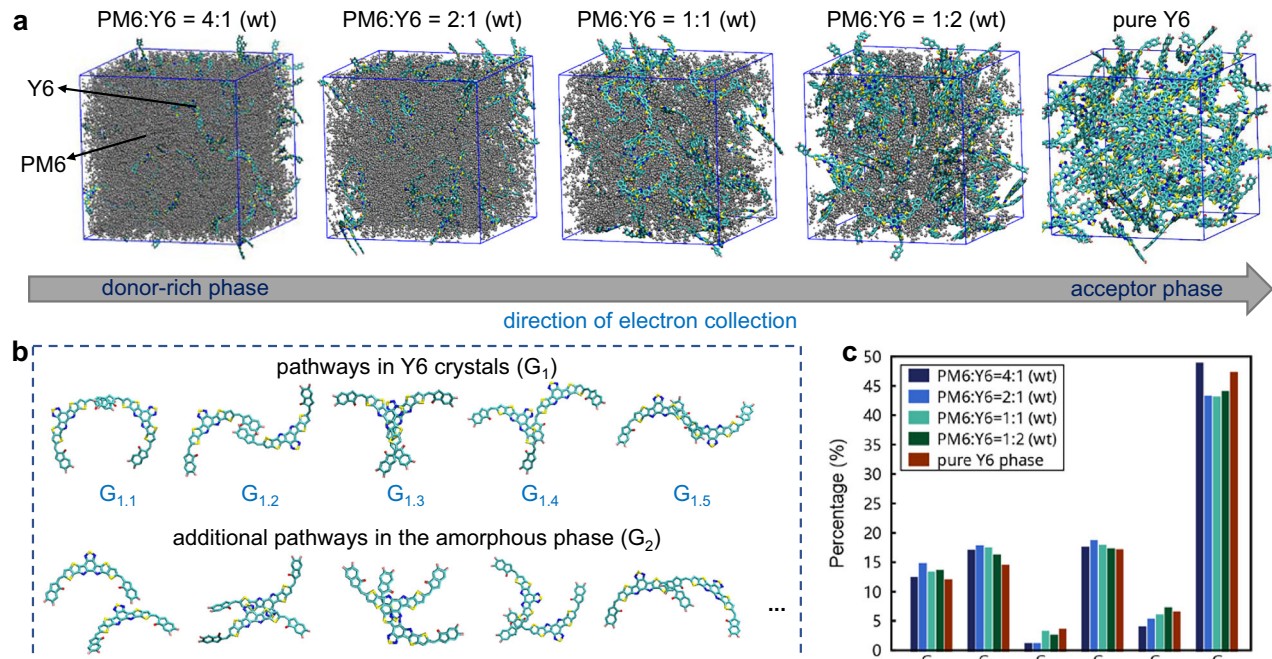

**Fig. 6 | Illustration of the application of the HiDiscover protocol to study the electron transport pathways in different regions of the PM6:Y6 bulk heterojunction. a** Molecular models as a function of Y6 ratio (the Y6 side chains are not shown, and the PM6 chains are displayed in gray). **b** Short-range Y6-Y6 contacts identified in the crystalline region and the ones that appear in the amorphous domains and are not present in the crystalline domains. **c** Ratios of short-range Y6-Y6 contacts (among all contacts) in the PM6:Y6 blend, showing as percentages.

Notably, in both the mixed and acceptor domains, around 50% of short-range Y6-Y6 contacts resemble those observed in the Y6 single crystals (Fig. 6b, c). At PM6:Y6 ratios from 4:1 to 1:2 (wt), the portion of crystalline-like molecular contacts is even higher than in the pure Y6 phase (Fig. 6c). Specifically, tail-to-tail ($G_{1.1}$ and $G_{1.2}$) and tilted ($G_{1.4}$) Y6-Y6 interactions account for approximately 50% of these short-range contacts. Compared to the disordered Y6-Y6 configurations found only within the amorphous packing ($G_2$, see Fig. 6b), these crystalline-like configurations exhibit well-overlapped π-conjugation areas, which can lead to efficient electron transport pathways throughout the mixed and acceptor domains, thereby contributing to the high performance of the Y6 acceptor. Interestingly, the configuration with the highest overlap, $G_{1.3}$, is less prevalent than $G_{1.1}$, $G_{1.2}$, and $G_{1.4}$, particularly in donor-rich domains. This points to opportunities for improving electron transport efficiencies through further molecular engineering. These findings underscore the value of probing the detailed microscopic features in complex organic solar-cell active layers and pave the way for discovering optimal molecular configurations in more complex ternary systems in which an additional donor or acceptor component is present and which lead to the highest reported power conversion efficiencies, now over 20% for organic photovoltaics[67,68].

## Discussion

Machine learning approaches have considerably impacted materials research in recent years. Notably, machine learning force fields have been developed that allow for simulating larger systems with accuracies approaching ab initio methods at reduced computational cost[69,70]. These methods have greatly expanded materials energy and environmental research, which often deals with complex systems that call for the modeling of large systems. However, an equally important task lies in apprehending the results from the modeling of the complex systems. While ML-based approaches have been greatly advanced to analyze molecular simulation data[27,43,45,71–73], general methods to identify meaningful atomic/molecular arrangements

within complex, multi-component materials frequently encountered in energy and environmental research have not been explored. This work establishes effective methodologies to identify meaningful patterns of atomic and molecular arrangements in these advanced materials, addressing a major challenge in their microscopic mechanistic studies. Specifically, we have developed a research protocol named HiDiscover and enabled the systematic extraction of microscopic configurational information in intricate molecular systems, which is crucial to mechanistic understandings of materials to inform their design, but challenging to probe by conventional analysis. This approach is particularly suitable for investigating multi-component systems with various interfaces frequently encountered in advanced materials, energy and environmental applications, as we demonstrated in the cases of Li-ion transport in a covalent organic framework, gas adsorption in a metal-organic framework, and molecular packing in organic photovoltaics.

In the context of Li-ion transport within 2D COFs, prior investigations have largely been empirical, constrained by a lack of methods capable of probing detailed microscopic conformations. This limitation has impeded progress in optimizing solid electrolytes. Our approach addresses this challenge, revealing detailed features of Li-ion transport in COF-PEO-3. These offer insights crucial for guiding the future design of 2D COFs aimed at enhancing ionic conductivity. Regarding gas adsorption in MOFs, our innovative approach facilitated the characterization of $CO_2$ transport pathways within MOF-5, which represent important features for designing materials for molecular separation applications, yet overlooked by previous simulations. Finally, using the HiDiscover protocol, we have quantitatively characterized the molecular packing of the Y6 acceptor across various domains in organic solar cells, which is a critical determinant of electron transport efficiency across long distances but remains understudied due to its inherent complexity. These findings highlight the potential of our method to advance material research through the integration of machine intelligence.

By applying the HiDiscover protocol, we can derive quantitative microscopic features from exhaustive machine-learned interpretations in a standardized way. Compared to the conventional practice that heavily relies on researchers directly analyzing original molecular simulation data (often in a customized way influenced by the researcher's experience), the HiDiscover protocol also mitigates the issue of incomplete observations made by individuals. Consequently, it minimizes bias and enhances the reliability of the resulting mechanisms.

We note that the HiDiscover protocol can be used with common molecular simulation techniques, making it applicable to the study of a wide range of materials and shifting computational materials research from qualitative and isolated observations towards more quantitative and comprehensive descriptions. At this stage, we anticipate no impediments to the application of the HiDiscover protocol to other complex, multi-component materials beyond those discussed here. In combination with already well-established molecular simulation tools, it can significantly contribute to advancing our understanding of microscopic mechanisms in materials at the atomic and molecular levels, thereby accelerating the design and optimization of advanced materials.

As discussed above, the objective of HiDiscover is to identify meaningful atomic and molecular arrangements within complex, multi-component materials. To the best of our knowledge, no general methodology, whether ML-based or otherwise, currently exists to address this complexity. It is useful here to note some of the design considerations as well as limitations of the HiDiscover protocol for its application and future development:

(1) Since our focus is on identifying atomic and molecular arrangements (static features), the temporal correlation in the MD trajectory is not treated in the model training. Instead, the simulated frames are treated as uncorrelated samplings of the systems around their equilibrium states. The temporal correlation in the dataset is generally weak and has very little impact on the results (see Section 4.4 in the SI). Although the temporal correlation in the data is not learned, mechanisms pertaining to temporal evolution can still be inferred by mapping predicted labels onto MD trajectories for analysis, as exemplified in our study on Li-ion transport.

(2) To most effectively use the HiDiscover protocol, a reasonable prior knowledge of the studied material is required. It is worth stressing that this prerequisite is not unique to HiDiscover but is implicitly present in traditional mechanistic studies of materials through molecular modeling as well, albeit often unacknowledged. Here, we emphasize that insights gained from existing research, preliminary simulations, and even human observations can facilitate the use of the HiDiscover protocol as well. In practice, applying the HiDiscover protocol does not seem to substantially increase the effort required to obtain prior knowledge, as demonstrated in three distinct examples.

(3) It is important to note that the HiDiscover protocol does not automatically identify the required tasks and relies on user guidance instead, similar to the interest points of a researcher in a conventional MD study. In a conventional MD study, the researcher may observe the data and adjust the interest points as the study is performed. Similarly, the tasks in a HiDiscover protocol may also be tuned if the researcher gathers new information (e.g., from new research, preliminary observations, and analysis). We emphasize that explicitly defining the tasks instead of implicitly using the interest points of the researchers is expected to improve the reproducibility of material mechanistic studies. A fact in materials modeling is that, while the calculated properties (e.g., mobilities, diffusivities) are often reproducible across different publications, the proposed mechanisms for the same material are often highly researcher-dependent as researchers typically focus on different aspects based on their experience (a fact that may not be documented along with the discovered mechanisms). In a HiDiscover protocol, the tasks are to be explicitly provided along with the discovered mechanisms, thus promoting the reproducibility of materials mechanistic studies.

(4) Reference molecular models are also frequently involved in conventional molecular dynamics studies. In the HiDiscover protocol, we take advantage of a set of reference systems for machine learning, allowing us to derive quantitative features of the atomic and molecular arrangements. Given the diversity of multi-component materials in energy and environmental research and the broad range of the mechanisms of interest, the methods to design the reference models are also expected to be versatile. For studies using molecular modeling, determining the reference systems is often obvious to the researcher since the objective is well-defined and a good prior understanding has been (or will be) obtained. Here, we have discussed four different types of materials and carried out calculations for three of them, demonstrating the feasibility of designing reference systems in the context of mechanistic studies of specific materials. These examples encompass solutions, porous composite materials, and organic heterojunctions at varying concentrations and degrees of mixing, which collectively serve as a reference for applying the HiDiscover protocol to other types of multi-component materials. A rule of thumb in the design of reference systems is to use simpler molecular models comprised of partial components of the target multi-component material, which are generally easier to construct and share similarities to the studied material. To aid this process, we have provided reference system suggestions for general multi-component materials (see Section 6 and Table S4 in the SI).

(5) In the HiDiscover protocol, the reference systems do not exclude the target system. For example, if our focus is on the impact of concentration on the molecular arrangements, a series of target systems of varying concentrations would be built even in a conventional MD study, which naturally becomes the complete reference systems in the HiDiscover protocol (as in task $\zeta_F$). In other cases, it is still suggested to set the target system (or representative ones from the target systems) in the reference systems to include all pertinent atomic/ molecular arrangements as done in this study, except when the researcher is confident that all pertinent features in all the target systems have been included in the reference systems. An example would be minor variations of the target systems (e.g., small concentration or temperature changes), which are unlikely to introduce significant new atomic/molecular arrangements and thus only a single representative target system may be included in the reference systems.

(6) If we have included the target system in the reference systems and only a small portion of predicted labels correspond to the other reference molecular systems, we will know that the selected reference systems do not contain the majority of the atomic and molecular arrangements present in the target multi-component material. This result tells us how different the target system is from the remaining reference systems, which in itself can also be useful information depending on the researcher's goal. In this case, all features of atomic and molecular arrangements in the multi-component material are still captured, although they cannot be linked to the other reference systems for easy human interpretation. Optimizing the list of reference molecular models may be desirable for the researcher, similar to the process of designing molecular models iteratively often done in a conventional MD study.

(7) The HiDiscover protocol may introduce new molecular models when the target systems being studied do not cover all reference systems. To sample sufficient atomic/molecular arrangements, parallel MD simulations with different initial configurations are recommended for the reference systems. Long MD simulations are also preferable. These are expected to increase the computational cost compared to only simulating the target systems in a conventional MD approach. In this study, we have tried to run multiple long MD simulations for the reference systems within our computational resources. Importantly, depending on the studied problem, using the HiDiscover protocol only

leads to 0–150% increase in the computational time compared to studying the same problems with conventional MD simulations (that cannot yield the same quantitative insights as in the HiDiscover protocol). Detailed timing information can be found in Section 4 of the SI. In general, the increased computational time when using the HiDiscover protocol compared to conventional MD study is related to the cost associated with new molecular models. Very interestingly, we have also performed tests on the dataset size for the reference systems and found that using just the first 1% of the MD production runs can also achieve robust results (see also Section 4 of the SI).

(8) To derive reliable material mechanisms, long MD simulations are usually needed for the target systems, for reasons similar to that discussed above. In this respect, using HiDiscover or a conventional protocol is not different. As such, the researchers still need to pay attention to the simulations on the target systems when using HiDiscover to investigate materials mechanisms.

## Methods
### Design of the contexts for Li-ion transporting in COF-PEO-3
Within the composite system of COF-PEO-3 electrolyte, multiple phases coexist, leading to the formation of various interfaces. Our previous investigation indicated that salt aggregates into clusters within the 2D COF pores[53]. These clusters can be in contact with the COF framework or isolated from other components. Building upon this understanding, we differentiate in task $\zeta_A$ between $Li^+$-$ClO_4^-$ configurations found in crystalline $LiClO_4$ ($A_1$), amorphous $LiClO_4$ ($A_2$), at the surfaces of $LiClO_4$ clusters ($A_3$), those emerging when $LiClO_4$ is blended with $(PEO)_3$ ($A_4$), those near the COF framework ($A_5$), and the additional configurations arising in the target system ($A_6$).

As a prior study has suggested different ion transport pathways at varying distances from the 2D COF framework[47], we distinguish in tasks $\zeta_B$ and $\zeta_C$ between ions near the 2D COF framework ($B_1$ and $C_1$), those within the pores ($B_2$ and $C_2$), and new configurations emerging in the target system ($B_3$ and $C_3$). Regarding task $\zeta_D$, we differentiate between the configurations of the $Li^+$-PEO chain in bulk $(PEO)_3$ ($D_1$) and the new ones emerging in COF-PEO-3 ($D_2$), recognizing that the framework imposes restrictions on the PEO chains and potentially influences their arrangements around the Li-ion.

### Molecular dynamics simulations
**COF-PEO-3.** We employed the all-atom optimized potentials for liquid simulations (OPLS-AA) force field[74], which has been widely applied in 2D COF studies for its efficiency and accuracy[75–79]. MD model 1 corresponds to an $8 \times 6 \times 5$ supercell of $LiClO_4$. MD model 2 contains 1000 $LiClO_4$ molecules randomly placed in a box with an initial size of $4.5 \times 4.5 \times 4.5$ nm. MD model 3 is a $LiClO_4$ cluster, built from the final molecular structure of MD model 2, with the new box size set to $9 \times 9 \times 9$ nm. MD models 4–6 contain 60, 30, and 240 $LiClO_4$ molecules (corresponding to $Li^+$ ratios of 0.46 wt%, 0.24 wt%, and 1.5 wt%, respectively) mixed in $2 \times 2 \times 20$ supercells of COF-42, respectively. MD model 7 contains 60 $LiClO_4$ and 480 $(PEO)_3$ in a box with an initial size of $6 \times 6 \times 6$ nm. In MD model 8, 360 $LiClO_4$ were initially randomly blended in a $2 \times 2 \times 20$ supercell of COF-PEO-3. The ratio of $Li^+$ is 1.4 wt%, close to that used experimentally[43]. Four parallel structures were constructed for MD models 2–8. Details can be found in Section 2 of the SI.

**MOF-5.** We turned to the Universal force field, which has been widely employed in studying gas adsorption in MOFs[59,60,80]. In MD models 9–12, 1000 $CO_2$ molecules were initially randomly placed in a box; pressures of 1, 10, 100, and 1000 bars were applied, respectively. In MD models 13-16, 25, 99, 493, and 985 $CO_2$ were initially randomly put in a $2 \times 2 \times 2$ supercell of MOF-5; these correspond to concentrations of 0.5 mmol/g, 2 mmol/g, 10 mmol/g, and 30 mmol/g, respectively. Since these systems are relatively well mixed, we use a single initial structure for each MD model. Details can be found in Section 2 of the SI.

**PM6:Y6 heterojunction.** We used the OPLS-AA force field, which has been extensively considered for the description of organic solar-cell active layers[74,81,82]. MD models of crystalline Y6, amorphous Y6, and PM6:Y6 with different weight ratios were constructed. In MD model 17, a $2 \times 1 \times 2$ supercell of Y6 was constructed based on its crystal structure[49]. This model is used to learn the short-range contacts in Y6 crystals. In MD model 18, 100 Y6 molecules were initially randomly placed in a box; this model represents the pure acceptor phase. In MD models 19–22, 100 Y6 and 3, 6, 12, or 24 PM6 chains consisting of 20 repeat units were initially randomly placed in a box; these represent regions going from an acceptor-rich domain to a donor-rich domain. Sixteen parallel structures were constructed for each MD model except for Model 17. Details can be found in Section 2 of the SI.

**Simulation details.** Unless otherwise mentioned, each MD model underwent an initial energy minimization (steepest descend algorithm), followed by equilibrium and production runs. The short-range electrostatic and van der Waals cutoffs were set to 1.4 nm. The smooth particle-mesh Ewald (PME) method was used for long-range electrostatic interactions[83]. A velocity rescaling scheme was considered for thermostat[84] and Berendsen, for barostat[85]. The time step was set to 1 fs. During the production run, structures were output every 10 ps. All molecular dynamics simulations were performed using the GROMACS package (version 2021.5)[86].

**Notes on the molecular models.** To ensure the robustness of the incremental learning model, it is necessary that the molecular dynamics data used in the preceding training step encompass the relevant ionic arrangements for the current class-incremental training. For example, MD model 1 should generate a substantial amount of ionic arrangements to include those belonging to context $A_1$ as observed in MD models 2, 3, 4, 7, and 8. This point should be given special attention in the case of Li-ion transport in COF-PEO-3 that exhibits inhomogeneous mixing. The dataset also needs to be sufficiently large to capture all significant ionic arrangements relevant to its context in the subsequent training as well. This can be achieved by conducting multiple long MD simulations on well-equilibrated systems with sufficient sampling. The sizes of the datasets range from ~$10^5$ to $10^7$, as listed in Table S2. Tests on the size of the dataset can be found in Section 4 of the SI.

**Notes on the relevance of the contexts.** We acknowledge that the depiction of microscopic mechanisms inherently involves a certain degree of subjectivity. When analyzing the atomic/molecular arrangements in molecular simulation trajectories, human interpretations heavily rely on intuition and experience. In previous machine learning models designed to classify atomic arrangements, these interpretations corresponded to the data labels, often categorized as solids, solid-liquid interface, fcc-type crystal, etc. Within our incremental learning approach, the defined context set introduces language elements through which machine-learned features are expressed. Ideally, these defined contexts should align with human comprehension, making them easily understandable. In practical terms, the context set can relate to the attributes of simpler molecular systems, establishing connections to the complex problem at hand. This approach provides a flexible method for introducing high-resolution labels and allows for detailed differentiation of intricate components and interfaces in the target system; this facilitates the extraction of deep insights from the simulation data and a more comprehensive description of the microscopic mechanisms. The primary limitation in defining the contexts lies in the feasibility of constructing the necessary dataset from molecular modeling.

## Data processing

**COF-PEO-3.** To process the raw data obtained from molecular dynamics simulations, we extract the local environment of each $Li^+$, represented by the coordinates of selected groups of atoms surrounding it. Considering that neighboring species often have identical atoms (e.g., the four oxygen atoms in $ClO_4^-$), as each coordinating with the ion would lead to similar configurations, we define equivalent atom groups to encompass multiple atoms in a molecule. The closest of the atoms to the ion is used to calculate the relative position of an equivalent atom group. This also reduces the record length for easier model training. Additionally, symmetry and sort operations were performed. These relative positions are subsequently converted into a Coulomb matrix format to generate the dataset. The dataset is then divided into training, validation, and test sets at an 8:1:1 ratio. Further information regarding the data processing procedure can be found in Section 3 of the SI.

**MOF-5 and PM6:Y6 heterojunction.** The data processing procedure is similar to that used for COF-PEO-3. For MOF-5, we focus on the configurations among $CO_2$ molecules and the arrangement of $CO_2$ with respect to the metal oxide core and phenyl ring linker. For PM6:Y6, we focus on the short-range Y6-Y6 contacts. Detailed information can be found in Section 3 of the SI.

## Model training

**COF-PEO-3.** We first train the class-incremental model within the inner layer of the hierarchical incremental learning framework. As shown in Fig. 3a, we initially applied unsupervised learning on the training set from MD model 1, employing k-means clustering, which offers a high level of interpretability. This resulted in classification model 1 with centroids corresponding to $Li^+$-$ClO_4^-$ configurations found in $LiClO_4$ crystals. Subsequently, the training set derived from MD model 2 was employed for class-incremental learning using a modified k-means algorithm, leading to classification model 2. This particular model possesses the capability to distinguish the context of $Li^+$-$ClO_4^-$ between crystalline and amorphous states. Then, four additional iterations of class-incremental learning were sequentially performed, using the training sets obtained from MD models 3, 7, 4, and 8. This ultimately leads to classification model 6 for task $\zeta_A$ ($\mathbf{x}_A \rightarrow A \times y_A$), gaining the new ability to identify $Li^+$-$ClO_4^-$ configurations on cluster surfaces, within interfacial regions in the PEO-chain environment, or next to the 2D COF framework, as well as those emerging in COF-PEO-3.

For $Li^+$-COF configurations, the datasets corresponding to MD models 4, 5, and 8 are used. Unsupervised learning was employed to generate classification models 7 and 10 using the training sets from MD model 4. Subsequently, the training sets obtained from MD models 5 and 8 were utilized for class-incremental learning, eventually leading to classification models 9 and 12 for tasks $\zeta_B$ ($\mathbf{x}_B \rightarrow B \times y_B$) and $\zeta_C$ ($\mathbf{x}_C \rightarrow C \times y_C$), respectively. Regarding $Li^+$-PEO configurations (tasks $\zeta_D$), a similar process of incremental learning is conducted using datasets derived from MD models 7 and 8. Consequently, classification model 14 is obtained ($\mathbf{x}_D \rightarrow D \times y_D$), effectively characterizing the configurations related to the $Li^+$-PEO side-chain interactions in the target system. Conducting sequential training of all the inner class-incremental learning models corresponds to task-incremental learning in the outer layer.

Details of the training processes and the determination of the $k$ values can be found in Section 4.1 of the SI. Accuracies of > 99% are achieved in our incremental learning model.

**MOF-5 and PM6:Y6 heterojunction.** The model training procedure is similar to that for COF-PEO-3. Accuracies of >99% are again achieved in our incremental learning model. Details can be found in Sections 4.2 and 4.3 of the SI.

**Notes on the sequence of datasets.** The order of the contexts $\Omega$ fed into the model is a factor in determining the labels. There are four possible scenarios for two successive instances $\Omega_i$ and $\Omega_j$: (i) When there is no overlap in context between $\Omega_i$ and $\Omega_j$, the situation corresponds to the conventional case of acquiring labeled data. In this instance, switching the order of $\Omega_i$ and $\Omega_j$ will have no impact. (ii) If $\Omega_i$ is a subset of $\Omega_j$, the smaller subset should be considered first, otherwise it will be represented by the larger dataset and thus will not contribute to determining the labels in the target system. (iii) When $\Omega_i$ and $\Omega_j$ have overlapping contexts but also contain parts exclusive to each other, the overlapped context will be incorporated into the label associated with the first dataset. (iv) $\Omega_i$ and $\Omega_j$ may also have identical contexts, in which case only one is needed. In general, it is desired to gradually increase the complexity of the datasets. In practice, to determine $\Omega$ and their order, researchers can rely on their background knowledge of the complexity of the molecular systems and verify it during model training. For instance, in case (ii), reversing the order would lead to the second label having minimal data coverage, thereby providing little assistance in distinguishing the atomic and molecular arrangements; in case (iv), adding $\Omega_i$ and $\Omega_j$ together would lead to severely reduced accuracy of the model. Detailed considerations for the determination of the dataset order in the this work can be found in Section 4 of the SI.

## Analysis of MD data with machine-generated interpretations

By applying the trained model to the data obtained from an MD trajectory of the target system, one can obtain a temporal sequence of labels that depict the evolution of microscopic states. Specifically, for COF-PEO-3, the final classification model was employed on the dataset derived from the MD simulations of COF-PEO-3 filled with $LiClO_4$, generating the labels for each $Li^+$ with various components at different frames. For MOF-5, the final classification model was employed on the MD systems with different $CO_2$ loadings. For the PM6:Y6 heterojunction, the final classification model was applied to the MD systems with different PM6:Y6 ratios. We then perform analysis on these machine-learned labels, reducing the reliance on directly analyzing the raw MD data as done in a conventional study routine. In particular, we calculated the ratio of each label in the target system and their correlation coefficient. In the time sequence analysis, to suppress the impact of errors from the machine-learning model, we use a criterion of 3 or more successive occurrences (corresponding to a 30-ps time window) of the same (or different) states to determine the start (or end) of a state.

## Reporting summary

Further information on research design is available in the Nature Portfolio Reporting Summary linked to this article.

## Data availability

Source data are provided with this paper. The complete datasets generated in this study exceed 400 GB and are available from the authors upon request, with responses generally provided within two weeks. Uniformly sampled subsets of the datasets have been deposited in the Zenodo database under accession code https://doi.org/10.5281/zenodo.132928010 and in Figshare under accession code https://doi.org/10.6084/m9.figshare.29995318. The initial and final configurations of the molecular dynamics trajectories in this study have been deposited in the Zenodo database under accession code https://doi.org/10.5281/zenodo.16880925. Source data are provided with this paper.

## Code availability

The code for model training is under Apache License (Version 2.0) and is available on GitHub at https://github.com/ShuSmeMat/HiDiscover or Zenodo at https://doi.org/10.5281/zenodo.16882233[87].

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

## Acknowledgements

H.L. acknowledges support by the National Natural Science Foundation of China (grant numbers: 22473072 and 22103053) and the Shanghai Technical Service Center of Science and Engineering Computing at Shanghai University. N.L. acknowledges support by the National Science Foundation Grant DMS-2418979. W.T. acknowledges support by the National Key R&D Program of China (grant number: 2023YFB3003001) and the Xiaomi Young Scholar Project. J.-L.B. acknowledges support by the Office of Naval Research, Award No. N00014-24-1-2114. The authors are also grateful to the University of Arizona Institute of Energy Solutions and Office for Research, Innovation, and Impact for support via the Arizona Technology and Research Initiative Fund. The computational work was supported in part by a grant of computer time from the DoD High Performance Modernization Program.

## Author contributions

H.Y. Zhang and H.Y. Li conceived the idea, conducted molecular dynamics simulations, wrote the code, analyzed the data, and prepared the initial draft. H.Y. Zhang, N. Lin, and H.Y. Li developed the methodology and reviewed and edited the manuscript. S.M. Pratik and T.H. Wang helped with the molecular simulations and contributed to writing the manuscript. A.M. Evans and J.-L. Bredas participated in the analysis of the data and reviewed and edited the manuscript.

## Competing interests

A Chinese patent (CN 202410963353.5) on the algorithms for training models and predicting labels was filed, listing H.Z. and H.L. as inventors. The remaining authors declare no competing interests.
