## [Transparent Peer Review file · Nature Communications]

Hierarchical incremental learning deciphers molecular arrangements in multi-component materials

Corresponding Author: Professor Haoyuan Li

Version 0:

Reviewer comments:

Reviewer #1

(Remarks to the Author)

The manuscript presents HiDiscover, a framework designed to identify human-interpretable atomic and molecular arrangements in multi-component systems using data from molecular dynamics (MD) simulations. This approach could offer valuable insights into molecular mechanisms for materials design and the authors applied this method to three different systems.

Although an interesting approach, I have several concerns about the framework as outlined below.

1. My primary concern with a machine-learning (ML) based method like this is the attempt to predict atomic/molecular mechanisms using MD trajectories without treating the data as a time series. When investigating system dynamics e.g., atomic interactions (which is inherently time-dependent), bypassing such crucial treatment in the ML algorithm will likely result in incorrect conclusions.
2. The method appears to require significant a priori information about the system under study, e.g., it requires reference molecular models representing relevant arrangements. For many practical systems, these arrangements are unknown a priori as acknowledged by the authors. Furthermore, due to this reliance on predefined molecular arrangements, HiDiscover will be unable to identify novel patterns absent in the list of candidate/"reference" models.
3. The framework depends on extensive MD simulations for both reference and target systems. The challenge of achieving converged sampling in MD simulations is well-documented, and long production runs could require prohibitive simulation times for some systems. A systematic demonstration of the effects of MD length on the obtained results should be provided.
4. Some steps described in the manuscript, particularly in Section 2.1, lack the level of detail necessary to be actionable. For example, the statement "reference molecular models should have similarities to the target system while demonstrating diversities to allow a fine differentiation of microscopic configurations" is too vague and a practical strategy should be provided that users can follow to test if this requirement is satisfied.
5. Positioning with respect to existing literature is almost non-existent. A vast number of ML based methods aimed at analyzing MD trajectories have been developed in the last few years. Discussions of such methods will improve the manuscript's readability.
6. As this is a "method development" work, adequate benchmarks e.g., against conventional non-ML methods should be provided. In this regard, experimental validation of the identified arrangements by HiDiscover can also be helpful.
7. HiDiscover protocol forces the final label to collapse into a reference label without providing an appropriate confidence metric on its prediction. To avoid the risk of generating unreliable labels, a probabilistic model/module capable of providing an uncertainty measure should be adopted.

(Remarks on code availability)

Reviewer #2

(Remarks to the Author)

In this study, the authors introduced a hierarchical incremental learning protocol to systematically explore multi-component materials. With this approach, they investigated Li-ion transport and gas adsorption in nanoporous materials, as well as molecular packing in organic active layers for photovoltaics. They asserted that this protocol addresses the problem of

incomplete observations typically made by individuals, compared to conventional methods that heavily depend on researchers. The development of a method to determine atomic structure using the incremental learning protocol, free from researcher bias, is highly commendable. On the other hand, there are some unclear points in the results and discussion. I would appreciate it if the authors could kindly revise the manuscript based on the comments below.

1) In page 24, the authors state "the HiDiscover protocol also mitigates the issue of incomplete observations made by individuals. Consequently, it minimizes bias and enhance the reliability of the resulting mechanism", which would be the selling point of this study. However, it is unclear how the authors choose the four tasks, zeta_A, zeta_B, zeta_C, and zeta_D, for Li-ion transport in COF-FEO-3 on page 11. To a reviewer (and likely many readers) unfamiliar with this material, it appears that the selection of these tasks may involve some prior knowledge or bias. A more detailed explanation would be appreciated to better illustrate the benefits of this study for a broader audience.

2) In the end of page 2 (abstract), "More broadly, this report highlights the potential of incremental learning to provide atomistic insight into complex materials and processes." I would like to understand the scope of this protocol's application. If a reader is, for instance, interested in high-entropy alloys as an example of complex materials, can this protocol handle complex compositions, structures, and properties in a straightforward manner? If so, can the results be reproduced using only the information provided in this paper and the supplementary information? Or is it necessary for the authors to develop additional code or tools to achieve this? If it is the former, this research could have a significant impact on related fields. However, if it is the latter, while useful, its impact on other areas would be more limited. I believe that discussing the versatility of the protocol is also important in assessing the overall usefulness of this paper. I would appreciate it if the authors could comment on this point.

3) In page 8 of Supplementary Information, the authors state "We lower the computational cost using a heuristic sequential approach." Could you provide details on the computational cost involved in deriving the results of this study? I would like to know the computation time and the computing resources used, such as the number of CPU cores (or GPUs).

(Remarks on code availability)

The code provide a readme file with enough instructions for running the application.

Version 1:

Reviewer comments:

Reviewer #1

(Remarks to the Author)

I thank the authors for their response and the revised manuscript. However, I find that my primary concerns were not adequately addressed, and the core issues remain unresolved. My specific concerns are outlined below:

1. HiDiscover is framed as a tool for uncovering structural arrangements relevant in understanding molecular mechanisms from MD data. However, in my opinion, there is absolutely no basis for HiDiscover to learn mechanistic insights for any dynamical processes, simply because it does not consider the dynamics of the data (acknowledged by the authors in their response: "Since our focus is on identifying atomic and molecular arrangements (static features),..."). I raised this issue in my first round of review, to which the authors provided autocorrelation calculations. Unfortunately, this does not address the concern at all. I give an example to clarify what I mean: Imagine, there are two metastable states: (i) free CO₂, and (ii) adsorped CO₂. In the paper, the authors generated data from these two states and trained an ML model to comment on which features are relevant for the adsorption process (the training data included trajectories from more than two initial arrangements, but the idea behind this example stays true). However, to understand such mechanisms we need to look at the intermediate, transition state in-between these two states, where the relevant features can be completely different compared to which features are relevant in the two metastable states respectively. Thus, mechanistic insights, by definition, rely on the dynamic process connecting these states, and relevant features may arise only transiently during the transition. HiDiscover compares equilibrated configurations and draws conclusions that, in my view, do not meet the standards for mechanistic interpretation. This gets complicated since we typically don't know what/where is the transition state and its also short-lived which will likely get washed away and the system will collapse into a single metastable state when the authors do their equilibration+MD steps in HiDiscover.

2. I strongly disagree with the authors' claim that "In fact, the HiDiscover framework does not require additional prior information compared to a conventional MD study of multi-component materials..." In a conventional MD simulation, the dynamics and relevant configurations naturally emerge from first principles based on the equations of motion and an initial arrangement. In contrast, HiDiscover requires the user to supply additional reference arrangements, effectively directing the protocol toward specific features in the trajectory. This imposes a high level of system-specific prior knowledge and in my opinion reduces the method's general utility. I note that the other reviewer also raised similar concerns.

3. The authors state that they are exploring the feasibility of "substituting conventional MD study" with the HiDiscover framework. While such a capability would indeed be a significant advance, unfortunately the current manuscript offers no compelling evidence or results to support this claim.

(Remarks on code availability)

Reviewer #2

(Remarks to the Author)

I appreciate the authors' thorough revisions to the manuscript in response to the referees' comments. The authors have successfully addressed all the concerns raised, leading to significant improvements in the clarity and quality of the manuscript. The value of the HiDiscover protocol, particularly its ability to facilitate detailed differentiation and efficient extraction of ionic and molecular arrangements that are difficult to identify using conventional molecular simulations, is now clearly demonstrated. This conclusion is well supported by the presented results and discussions. Accordingly, I recommend the manuscript for publication in Nature Communications.

(Remarks on code availability)

The code provide a file with enough instructions for running the application.

Version 2:

Reviewer comments:

Reviewer #1

(Remarks to the Author)

I thank the authors for their response to my comments, which I appreciate. My final thoughts on the work are as follows:

From a theoretical/methodological standpoint, this work would have represented a significant advance if dynamic interpretations were generated by treating MD data as a time-series as I mentioned in the previous rounds. However, the authors clarified that they are aiming for static interpretations of molecular arrangements. Unfortunately, the field has become saturated with machine-learning driven static interpretation schemes, and in my view, this work for multi-component materials does not substantially advance the field to warrant broad impact, and the results are not fully convincing.

The proposed protocol involves many engineering steps that, as I noted in my initial reviews, are difficult to translate into actionable strategies for a user.

(Remarks on code availability)

Hierarchical incremental learning deciphers multi-component materials

Hanyin Zhang,^{a,b} Nan Lin,^c Austin M. Evans,^d Tonghui Wang,^e Saied Md Pratik,^f

Jean-Luc Bredas,^f and Haoyuan Li^{a,b}*

^aSchool of Microelectronics, Shanghai University, Shanghai 201800, China

^bKey Laboratory of Advanced Display and System Applications, Ministry of Education, Shanghai University, Shanghai 200072, China

^cDepartment of Statistics and Data Science, Washington University, St. Louis, MO 63130, USA

^dGeorge and Josephine Butler Polymer Laboratory, Department of Chemistry, University of Florida, Gainesville, Florida 32611-7200, United States

^eKey Laboratory of Automobile Materials, Ministry of Education, and School of Materials Science and Engineering, Jilin University, Changchun 130022, China

^fDepartment of Chemistry and Biochemistry, The University of Arizona, Tucson, Arizona 85721-0041, United States

*Corresponding author: lihaoyuan@shu.edu.cn

Abstract

Identifying meaningful patterns of atomic and molecular arrangements from molecular simulations is crucial for revealing microscopic mechanisms in materials. Unraveling these patterns is challenging for the multi-component systems frequently encountered in advanced materials, energy and environmental applications. This limits the understanding of the microscopic mechanisms that ultimately govern the performance of devices based on these systems. Here, we propose a hierarchical incremental learning research protocol named HiDiscover to systematically expedite the mechanistic exploration in multi-component materials. As illustrations, we study Li-ion transport and gas adsorption in nanoporous framework materials, as well as molecular packing in organic active layers for photovoltaics. The HiDiscover protocol enables the detailed differentiation and facile extraction of ionic and molecular arrangements, and reveals quantitative microscopic features that are difficult to discern through conventional molecular simulations, thereby informing materials design. Our approach is seen to improve the reliability of mechanistic descriptions for three different processes in three different classes of materials. More broadly, this report highlights the potential of incremental learning to provide atomistic insight into complex materials and processes.

1. Introduction

A key to developing novel active molecules, materials, or devices in materials, energy, and environmental sciences is the comprehension of the microscopic characteristics and temporal evolution of atomic and molecular systems.¹⁻⁶ At this point in time, mechanistic insight has been based largely on molecular simulations,⁷ which determine the evolution of atomic coordinates that are denoted as trajectories. In their raw format, these data do not directly inform any mechanistic process. Therefore, a crucial task for the researcher is to identify meaningful atomic/molecular arrangements, which are required to derive the microscopic mechanisms.⁸⁻¹⁰ However, gaining a clear understanding of the features of the often complex atomic/molecular arrangements within multi-component systems remains difficult, which limits their impact in terms of revealing the microscopic mechanisms.¹¹ This challenge is becoming increasingly significant as the research interests increasingly involve mixed organic/inorganic composites¹²⁻¹⁵ or interfaces¹⁶⁻¹⁹.

We take two-dimensional covalent organic frameworks (2D COFs) as an example, which have been extensively studied in recent years as solid-state electrolytes in metal batteries for increased safety and lower operating temperatures.²⁰⁻²⁵ Understanding the mechanisms underlying Li-ion transport in these frameworks is critical for their design towards higher ionic conductivities, yet our knowledge of the microscopic processes involved remains incomplete.²⁴ The challenge lies not only in the long timescales required to accurately capture the ion motion in the pores of 2D COFs,²⁴ but also in the lack of tools capable of extracting meaningful ionic arrangements from these composite and disordered systems. Currently, the interpretation of the molecular simulation data obtained on such systems heavily relies on a manual examination of trajectories (often incomplete due to the large

number of frames that need to be analyzed) and reasoning based on statistical indicators (such as the radial distribution functions) and the researcher's expertise.^{26–28} However, this approach generally leads to empirical mechanistic inferences instead of exhaustive and quantitative characteristics. The heavy reliance on the researcher's experience also introduces potential biases, which makes the resulting interpretations susceptible to subjective information.

Methods have been relatively well established to identify atomic arrangements in simple structures like crystals, with earlier approaches exploiting structural descriptors like the common neighbor analysis²⁹, centrosymmetry parameter³⁰, common neighbor parameter³¹, and others^{32–34}; these methods, however, often suffer from reduced accuracy at large deviations from the perfect crystal structures, **making them inapplicable to the complex, multi-component materials.**^{35,36} Recently developed machine-learning (ML) approaches have shown improved accuracy and broader applications to materials^{35,37,38} and have facilitated the descriptions of a few complex systems such as solid-liquid interfaces³⁹ and amorphous atomic solids^{40,41}. A key in **such approaches** is a dataset of labeled atomic structures built from a series of isolated reference molecular models that represent different types of atomic arrangements; **this dataset** is used to train the machine-learning model that then predicts the local structure within a target system by similarity. However, due to the inherent challenges in constructing datasets of isolated phases for multi-component materials with diverse interfaces, developing a machine-learning model capable of identifying atomic and molecular arrangements in complex systems remains a difficult proposition.^{38,39,42} **We note that, although machine-learning approaches have made significant advances in recent years for analyzing molecular simulation data, current**

methodologies remain inadequate for identifying meaningful atomic/molecular arrangements in complex multi-component materials.^{43–45}

Herein, we propose an incremental learning⁴⁶ approach to allow the use of overlapping datasets in training machine-learning models that identify atomic and molecular arrangements. This makes it feasible to construct reference molecular models for general multi-component materials. In practice, one can use a series of molecular models with similarities to the target system as a reference for machine learning. Our results demonstrate the feasibility of constructing these reference systems for diverse multi-component materials relevant to energy and environmental research applications. The labels can be appropriately defined and learned in an incremental manner by feeding datasets sequentially. We further divide the problem into separate tasks, each dealing with a portion of the whole system, facilitating model training while providing flexibility for focusing on the parts of interest. The predicted labels from different tasks are combined at a later stage, forming a complete descriptor of microscopic atomic and molecular arrangements. The final model has a hierarchical structure with task incremental learning in the outer layer and class incremental learning in the inner layer. As a result, this approach enables us to identify relevant atomic and molecular arrangements in general multi-component materials, addressing a critical challenge that remains intractable using previous methodologies.

In this framework, to systematically accelerate the mechanistic studies of materials, we have developed a research protocol that we named HiDiscover. This protocol allows the analysis of the machine-learned labels of atomic/molecular arrangements directly from the simulations, and reduces the reliance on a researcher's experience in addressing raw

simulation data often in a non-exhaustive way. To illustrate the efficacy of the HiDiscover protocol, we examine three distinct systems: (i) Li-ion transport in a 2D COF,³⁸ (ii) CO₂ adsorption in a metal-organic framework (MOF),⁴⁷ and (iii) molecular packing in the active layer of an organic solar cell.⁴⁸ Our approach brings forth quantitative microscopic insights into these complex systems, which cannot be obtained using conventional approaches. Thus, these representative examples demonstrate the potential of HiDiscover in accelerating mechanistic studies in the realm of materials, energy, and environmental sciences.

2. Results

2.1. The incremental learning approach

To discuss the problem in a general framework, we use $\{\mathbf{x}_i \in \mathbf{R}^d; i=1, 2, \dots, n\}$ to denote the space within which our system exists. The distribution of atomic and molecular arrangements typically displays prevailing patterns $\{y_i; i=1, 2, \dots, m\}$, which can be identified through techniques such as clustering.^{44,49} However, these results are often difficult to comprehend at the human level as we tend to describe atomic/molecular arrangements more effectively at an abstract level. For instance, we find it easier to comprehend the contrast between the arrangement of H₂O molecules in liquid and solid states, rather than memorizing the preferable degrees of orientation exhibited by H₂O molecules relative to one another.

With the description outlined above, we can categorize atomic patterns with similar meanings (contexts) to facilitate comprehension. Each context \mathcal{C} is a subset of $\{y_i\}$. For instance, when investigating the molecular arrangements of H₂O, we can define a context

set $\{\mathcal{C}_i; i=1,2,\dots,p\}$ consisting of $\mathcal{C}_{\text{liquid}}$ and $\mathcal{C}_{\text{solid}}$, which encompass patterns in the liquid and solid phases, respectively. These definitions align with the labels commonly employed for classifying atomic arrangements.^{11,35,39}

The main challenge when dealing with multi-component systems is that their molecular models often include contexts that are difficult to isolate, making the generation of datasets for individual labels challenging. Acknowledging this issue, we introduce Ω to represent a subset of $\{\mathcal{C}\}$. Then, it becomes feasible to construct molecular models that exhibit similarities to our target system, resulting in a series of context collections denoted by $\{\Omega_i; i=1, 2, \dots, q\}$. However, compared to the previous machine-learning approaches that construct non-overlapping datasets with known labels, Ω_i and Ω_j will likely overlap and thus have unknown labels, as illustrated in **Figure 1a**. As a result, these datasets cannot be directly used to train a machine-learning model. Here, we propose to adopt an incremental learning⁴⁶ methodology and feed these datasets sequentially into a model that learns the features in a step-wise manner. In this way, the resulting model discerns dissimilarities between different Ω contexts. Specifically, this class-incremental learning model identifies the excess context between successive datasets, which can serve as our label; the initial label corresponds to the first dataset and subsequent labels represent the excess contexts between successive datasets: $\{\Omega_1, \Omega_2-\Omega_1, \Omega_3-\Omega_2, \dots, \Omega_q-\Omega_{q-1}\}=\{A_1, A_2, A_3, \dots, A_q\}$. Such a machine-learning model can effectively predict labels in the target system. A further discussion on the relevance of the contexts can be found in **Section 4.1**. We note that the order of Ω fed into the model is a factor in determining the labels (see **Sections 4.2** and **4.4** for details). A more technical description of this approach is provided in **Section 1** of the **Supplementary Information (SI)**.

Figure 1. Illustration of the incremental approach. **a**, Collection of contexts. **b**, Hierarchical incremental learning framework. **c**, HiDiscover research protocol for simulating multi-component molecular systems.

A real-world problem often consists of multiple tasks (ζ), each with its specific context set $\{A\}$, $\{B\}$, $\{C\}$, ..., which may involve different data formats. For instance, consider a scenario where we want to investigate the molecular arrangements within an H_2O phase with itself and when a chemical compound (impurity) is mixed, which requires two tasks to be performed. To address such complexity, it is beneficial to incorporate a task-incremental learning model in the outer layer, as illustrated in **Figure 1b**. This results in a hierarchical incremental learning framework that is suitable for extracting features of atomic and molecular arrangements in multi-component molecular systems, facilitating a more comprehensive and accurate analysis. To promote its application in discovering microscopic mechanisms in materials, we have developed a research protocol named

HiDiscover (**Figure 1c**). The detailed steps and comparison with those in a conventional MD study are as follows:

- (i). Identify the material under study and determine the target system for modeling, which is similar to what is done in conventional MD studies. For example, here, we are interested in a meta-stable H₂O-impurity mixture at a given pressure.
- (ii). Acquire initial knowledge about the target system, which can be obtained from literature, prior research experience, or preliminary simulations. In a conventional MD study, the initial knowledge is necessary for the construction of all pertinent molecular models and will inform the researcher of “interest points” to observe and analyze from the simulation data. In the HiDiscover protocol, this knowledge will also inform the design of tasks and reference systems in the next step.
- (iii). Define tasks and design reference molecular models. This step in the HiDiscover framework differs from conventional protocols. Tasks correspond to specific “interest points” to which the researcher would pay attention during observation and analysis in a conventional study, though often implicitly and without documentation (compromising study reproducibility). Here, we explicitly define the tasks for training the machine-learning model. In conventional MD studies, reference molecular systems are also frequently involved, *e.g.*, to use their results as a baseline for comparison; yet, the information gained from this comparison by humans are often limited to intuitive observations and material properties (mobility, diffusivity, etc.) rather than detailed and quantitative atomic/molecular arrangements. When applying the HiDiscover protocol to multi-component materials, the reference molecular models can take the components

(such as phases and interfaces) present in the target system while considering the feasibility of model construction. In our example, we may have one task focusing on H₂O-H₂O configurations and the other on H₂O-impurity configurations. To better extract meaningful labels, the reference molecular models are suggested to have similarities to the target system while demonstrating diversities to allow a fine differentiation of the microscopic configurations. This may require good background knowledge of the studied material and some trial and error design, similar to the process of adding molecular systems for comparison in a conventional MD work. We note that, when a HiDiscover protocol is being considered instead of a conventional MD study, a good background knowledge of the studied material is already or will be obtained and a well-defined objective for the researcher to elucidate specific material mechanisms has been or will be established. As such, the ultimate design of the reference systems is feasible, as we demonstrate in several distinct examples. Simpler molecular models that incorporate partial components of the studied multi-component material are generally easier to construct than the target system when invoking the HiDiscover protocol (see **Section 6** and **Table S4** in the **SI** for further discussions). It is also suggested to incorporate the target system in the reference molecular models to capture potentially missed features. Here, we may build ice, liquid water, and H₂O gas as the reference models in the first task, while having a series of H₂O phases with different impurity concentrations in the second task.

- (iv). Construct the target and reference molecular models, as done in a conventional MD study. Multiple random initial structures need to be considered for inhomogeneous systems.
- (v). Perform molecular simulations for the reference and target molecular models, ideally with long production runs to achieve sufficient coverage of the atomic/molecular arrangements. This step and the considerations are the same as in a conventional MD study.
- (vi). Generate datasets based on the simulation data from the reference and target molecular models.
- (vii). Train each class-incremental model by sequentially feeding its datasets. In this example, we will train two sets of models. In the first task, we may feed datasets of ice, liquid water, and H₂O gas in an incremental learning model. In the second task, we can feed datasets of H₂O with increasing or decreasing impurity concentrations in the other set of models. We note that the assumed order of datasets may need to be tested for a fine differentiation of the contexts (see Notes in Section 4.2). Based on the final order of the datasets, the labels can be derived. Assuming the above-mentioned order of datasets, the three labels in the first task are H₂O-H₂O configurations in ice, additional H₂O-H₂O configurations in water, and additional H₂O-H₂O configurations in H₂O gas.
- (viii). Apply the trained model to the dataset derived from the target system, resulting in a sequence of labels and detailed features of atomic/molecular arrangements. In our example, we apply the two final classification models from the two tasks to the dataset of the target system. We then obtain two sets of labels

corresponding to each H₂O molecule in each frame of the simulated trajectory. We can also combine labels from the two tasks to form a complete descriptor of each H₂O molecule for further analysis.

- (ix). Analyze the characteristics of the machine-learned labels, reducing the reliance on researchers directly observing raw MD data.

Steps (vi)-(ix) are all different in the HiDiscover and conventional MD study. We now describe the application of this protocol to three distinct systems, illustrative of current, high-level research on multi-component materials used in energy and environmental applications.

2.2. Li-ion transport in COF-PEO-3

2.2.1. Model training

We first use the HiDiscover protocol to investigate Li-ion transport in COF-PEO-3, a representative 2D COF used as a solid-state electrolyte, as shown in **Figure 2**.⁵⁰ This COF is based on the COF-42 structure with a chemical grafting approach employed to incorporate poly(ethylene oxide) (PEO) chains, which has been shown to enhance Li-ion transport.⁵⁰⁻⁵² A critical step in understanding Li-ion transport in COF-PEO-3 is to know how the cations interact with the various species and their configurations. However, the amorphous nature of the salt-COF composite makes it challenging to identify the configurational features. Here, the HiDiscover protocol enables us to learn the distinctive characteristics of Li⁺ arrangements when the cations interact with COF-PEO-3 and the counter-ions ClO₄⁻. In the latter case, Li⁺ coordinates with the oxygen atoms of the perchlorate. Regarding COF-PEO-3, it is intuitive to distinguish between ionic

arrangements with the main COF framework and with the PEO side chains. Previous work has indicated that Li^+ forms stronger interactions with the more electronegative atoms,²⁴ specifically: (i) the nitrogen atoms in the COF linkers; (ii) the oxygen atoms at the COF-PEO connections; and (iii) the oxygen atoms in the side chains (**Figure 2**). Accordingly, we define four tasks, corresponding to $\text{Li}^+\text{-ClO}_4^-$ (ζ_A), $\text{Li}^+\text{-COF}$ framework linkage (ζ_B), $\text{Li}^+\text{-COF}$ side-chain connection (ζ_C), and $\text{Li}^+\text{-PEO}$ chain (ζ_D).

Figure 2. Illustrations of the reference and target molecular models, tasks, and labels in the HiDiscover protocol for studying Li-ion transport in COF-PEO-3. MD models 1-3 correspond to crystalline and amorphous LiClO_4 phases and a LiClO_4 cluster, respectively. MD models 4-6 correspond to LiClO_4 blended in COF-42 with various weight ratios. Structures with smaller weight ratios capture the arrangements close to the 2D COF framework while those with higher ratios explore the arrangements in the 2D COF pores. MD model 7 is LiClO_4 blended in pure (PEO)₃. MD model 8 is the target system. Circles

in the chemical structures highlight the regions in the 2D COFs and (PEO)₃ that are expected to interact strongly with Li⁺ (green, purple, blue, and brown circles correspond to tasks ζ_A , ζ_B , ζ_C , and ζ_D , respectively). Further information is provided in **Sections 2-3** of the **SI**.

We then design the reference molecular models for each task and determine the contexts, aiming to ensure configurational diversity, easy human comprehension, and the practical generation of datasets through molecular simulations (see **Figure 2** and **Section 4.1** for details). The chosen reference molecular models comprise crystalline LiClO₄, amorphous LiClO₄, a LiClO₄ cluster in vacuum, LiClO₄ in COF-42 (which we recall shares the same framework as COF-PEO-3 but lacks the side chains) with varying concentrations, LiClO₄ blended with PEO, and the target system LiClO₄ in COF-PEO-3. We note that the target system is set as the last one in the reference molecular models; thus, it is expected to capture the features missed by the previous reference molecular models. Details on the construction and simulation of these MD models, data processing, and model training can be found in **Methods** and **Sections 2-4** of the **SI**.

2.2.2. Discovered Mechanisms

The observations derived from the exhaustive machine-learned interpretations offer in-depth insights into the Li-ion configurations in 2D COFs, which would not be available via a conventional analysis of the MD simulations. Specifically, we can now quantitatively describe the ionic configurations in COF-PEO-3, expressed in terms of the labels derived from the reference molecular models we have designed. We first examine the main features of the Li⁺-ClO₄⁻ configurations (task ζ_A) in the COF-PEO-3 target system. **Figure 3a** shows the t-Distributed Stochastic Neighbor Embedding (t-SNE) plot of all Li⁺-ClO₄⁻

configurations and the ratios corresponding to different labels. As can be seen, there is negligible (0.2%) crystalline packing (A_1), while 1.4% of the total $\text{Li}^+\text{-ClO}_4^-$ configurations correspond to configurations in amorphous LiClO_4 , which display deformations from the crystal structure (see $A_{2.1}$ in **Figure 3a**). The configurations similar to those at the LiClO_4 cluster surfaces constitute 7% of the total $\text{Li}^+\text{-ClO}_4^-$ configurations, with oxygen atoms coordinating on one side of the Li-ion, *i.e.*, highly asymmetric ionic configurations (see $A_{3.1}$ in **Figure 3a**). Approximately 10% of the $\text{Li}^+\text{-ClO}_4^-$ configurations in COF-PEO-3 resemble those found in mixtures of LiClO_4 and PEO, typically with cations and anions separated by large distances (see $A_{4.1}$ in **Figure 3a**). Furthermore, 39% of the $\text{Li}^+\text{-ClO}_4^-$ configurations in COF-PEO-3 mirror the patterns observed when LiClO_4 is introduced into COF-42, involving 2-3 anions coordinated to the Li-ion (see $A_{5.1}\text{-}A_{5.3}$ in **Figure 3a**). Finally, 42% of $\text{Li}^+\text{-ClO}_4^-$ configurations exhibit new features in context A_6 (see $A_{6.1}\text{-}A_{6.2}$ in **Figure 3a**); these configurations demonstrate closer ionic proximity compared to A_4 but weaker coordination than in A_5 , which suggests that introducing the PEO side chains promotes ionic dissociation.

Figure 3. Predicted labels for the Li-ion configurations in COF-PEO-3 in the four tasks. The left panels in **a-d** provide a t-SNE visualization of the ionic configurations in tasks ζ_A - ζ_D ; labels in each task are differentiated according to color. The right panels in **a-d** show the ratios of different labels in each task and representative arrangements of Li-ions with various species for different labels (X_Y) and sub-labels ($X_{Y,Z}$). Pink, red, cyan, gray, and blue spheres represent Li, O, Cl, C, and N, respectively. Visual illustrations of the distributions of Li-ions in tasks ζ_B and ζ_C can be found in **Figure S112**.

The configurations of Li⁺-framework linkages (task ζ_B) and Li⁺-side-chain connections (task ζ_C) share similar trends. As **Figure 3bc** shows, approximately 40% of Li ions are in proximity to the main COF framework ($B_{1.1}$, $B_{1.2}$, $B_{1.3}$, $C_{1.1}$, and $C_{1.2}$), interacting with 1-2 functional groups, typically adopting an in-plane or between-plane configuration. $B_{2.1}$, $B_{2.2}$, $C_{2.1}$, and $C_{2.2}$, which account for ~50% of the configurations, feature a large separation from the main COF framework. The remaining 10% of Li-ion arrangements in COF-PEO-3 demonstrate an even larger separation from the main COF framework ($B_{3.1}$, $B_{3.2}$, $C_{3.1}$, $C_{3.2}$).

Regarding the Li⁺-PEO configurations (task ζ_D), approximately 24% of them in the target system correspond to context D₁; it displays two patterns, as illustrated in **Figure 3d**: D_{1.1}, with two PEO chains tightly locking the Li-ion; D_{1.2}, where one chain displays reduced interactions. The rest of the configurations (D_{2.1}, D_{2.2}, D_{2.3}) exhibit larger Li⁺-PEO distances, with the closest PEO chains either enveloping the Li-ion or positioned adjacent to it. This underscores the restrictions imposed by the COF framework on the PEO chains due to their covalent bonding, leading to the formation of a loosely bound PEO matrix for the Li ions as compared to the tighter binding found in salt-containing PEO electrolytes.

The combination of the contexts from the four tasks comprehensively depicts the microscopic states of the Li-ions. Overall, the prevailing state observed is (A₅,B₁,C₁,D₂), representing ~21% of all Li-ion arrangements (**Figure 4a**); it corresponds to Li ions close to the main COF framework and loosely bound to the PEO side chains. State (A₆,B₁,C₁,D₂), which has slightly weaker Li⁺-ClO₄⁻ interactions, accounts for roughly 12% of total arrangements. Three other states, (A₆,B₂,C₂,D₁), (A₅,B₂,C₂,D₂), and (A₆,B₂,C₂,D₂), each encompassing ~8%-14% of the total arrangements, correspond to the ions away from the main COF framework. Collectively, the five states we just discussed comprise ~65% of all Li-ion arrangements in COF-PEO-3.

Figure 4. Analysis of Li-ion transport in COF-PEO-3 based on machine-learned labels. **a**, Ratios of different microscopic states (combinations of the contexts from all four tasks) of Li-ions in COF-PEO-3 at 1.38 wt%. **b**, In-state mean square displacements (MSDs) of Li ions per unit time. The grey bars display the MSD value for all three directions in space; red and dark blue lines illustrate the values in the z and in-plane (x/y) directions respectively. **c**, Transition frequencies between two states derived from the MD simulations (4 random initial configurations, each with 1- μs simulation time). **d**, Number of Li-ions belonging to different contexts at different weight ratios of Li^+ in COF-PEO-3; 0.54 wt%, 1.00 wt%, and 1.38 wt% correspond to a total of 120, 240, and 360 Li^+ in the MD models. Refer to **Figure S123** for further information.

These complex features of the Li-ion configurations point to transport mechanisms distinctive from those found in traditional polymer electrolytes. By further analyzing the

time sequence of the machine-learned labels, we can get insight into the Li-ion dynamics. We computed the in-state mean square displacement (MSD) of the Li ions (**Figure 4b**) and the transition times between various states (**Figure 4c**). A detailed analysis is provided in **Section 5** of the **SI**. Overall, our analysis based on the machine-learned labels indicates that the Li-ion motion in COF-PEO-3 at room temperature is impacted by short-range interactions with the PEO side chains, affected indirectly by the 2D COF framework, and influenced by the Coulombic forces of the surrounding ions. This insight informs the future design of 2D COFs in terms of the following three aspects: (1) side-chain and decoration-group design to control the short-range motions of the Li-ions; (2) framework structure engineering that balances enhancing thermal vibrations that promote hopping among sites and anchoring functional groups that restrict long-range ion migration; and (3) the choice of Li salt (and other ionic groups) to adjust Coulombic interactions for efficient ionic movement along the pores.

We note that the quantitative assessment of the Li-ion configurations and motions discussed above correspond to COF-PEO-3 with 1.38 wt% Li⁺ (a value close to those used experimentally); the results will change as a function of the Li⁺ ratio. Notably, a significant ratio of ions is associated with contexts A₄ and D₁ at low Li-ion weight ratios (reaching probabilities of ~80% and ~60% at 0.54wt%, respectively), as shown in **Figure 4d**; these correspond to Li-ions well blended in the PEO matrix. We note that this quantitative insight has been overlooked in previous simulations of Li-ion transport in COFs.²⁴ Our machine-learning interpretations of the MD simulations point out that Li-ions prefer to bind at first to the PEO side chains when adding the Li salt to COF-PEO-3; adding more Li salt leads to additional Li-ions that loosely bind to PEO chains with non-uniform ionic configurations,

a process that can even reduce the number of Li-ions uniformly blended and tightly bound to PEO likely due to modifications to the PEO side-chain configurations. Thus, these results underline that it would be worth exploring experimentally the interactions of Li-ions with the various components of 2D COFs as their concentration increases. Furthermore, increasing the Li-ion concentrations leads to a reduction of the diffusivity, which indicates that there is a concentration with optimal ionic conductivity (**Figure S124**). Our work highlights that salt concentration is an intrinsic factor in Li-ion transport in 2D COFs, which needs to be considered in materials design instead of tuned empirically as an extrinsic factor, as done so far. In particular, the optimization of Li-ion transport at different salt concentrations should focus on their corresponding Li-ion-COF interactions.

2.3. CO₂ adsorption in MOF-5

Gas adsorption in metal-organic frameworks has been extensively studied, as it is closely related to separation, catalysis, sensing, and environmental applications.^{47,53-56} While molecular simulations have been useful in predicting diffusivities and adsorption capacities, the microscopic configurational features obtained from these studies, which are needed to fully unravel the adsorption and transport mechanisms, remain limited.^{53,57-62} Here, we demonstrate that using the HiDiscover protocol allows us to reveal in detail and quantitatively the arrangement features of CO₂ in the representative MOF-5. Specifically, we focus on understanding the features related to CO₂-CO₂ and CO₂-MOF-5 configurations, which will provide insight into CO₂ transport within the MOF. Such information is difficult to visualize by the eye as the gas molecules form dynamic and transient structures from which it is difficult to generate statistically meaningful knowledge. Here, the first task (labeled ζ_E) captures the features related to CO₂-CO₂ configurations (**Figure 5a**), using the

CO₂ gas at different pressures (1000-1 bar) as reference. The contexts E₁-E₄ correspond to CO₂ packings with densities of $\sim 1-0.003$ g/cm³. They act as density indicators and clearly depict the evolution of a denser CO₂ packing as the CO₂ loading into MOF-5 increases (**Figure 5b**). The second task (labeled ζ_F) describes the CO₂-MOF configurations: F₁-F₃ correspond to those that emerge at 0.5, 2, and 10 CO₂ mmol/g loadings, respectively.

F₁ is found to cover most of the CO₂-MOF configurations even at much higher CO₂ loadings. In other words, increasing the CO₂ loading does not lead to new CO₂-MOF configurations. A more detailed analysis shows F₁ has four features: F_{1.1} (in-pore corner configurations), F_{1.2} (off-corner configurations, forming pathways connecting adjacent pores), F_{1.3} (in-pore and between-face-center pathways), and F_{1.4} (through-face-center pathways and enveloping F_{1.3}), as illustrated in **Figure 5c**. We obtain that about half ($\sim 46-49\%$) of the CO₂ molecules are in off-corner configurations, suggesting that CO₂ prefers to transport around the metal oxide centers both in-pore and to adjacent pores. This result aligns with the human observation that CO₂ tends to coordinate with the Zn₄O₆⁺ cores.⁵⁵ Our further analysis reveals that the off-corner configurations can act as a bridge among the other three configurations during CO₂ transport (**Figure S125**). As the CO₂ loading increases, the ratio of molecules in off-corner configurations decreases while that in through-face pathways increases, suggesting that more CO₂ is transported to adjacent pores through the pore center (**Figure S126**). As such, the HiDiscover protocol enables a detailed differentiation of the molecular arrangements of CO₂ in MOF-5, which offers an insight not available from conventional molecular simulation protocols and can inform the molecular design of MOFs for gas adsorption, transport, and separation.

Figure 5. Illustration of the application of the HiDiscover protocol to CO₂ adsorption in MOF-5. **a**, Molecular models of CO₂ (10 bar) and MOF-5, and illustrations of the two tasks. **b**, Ratios of different contexts for 0.5, 2, 10, and 20 mmol/g CO₂ loading in MOF-5. Contexts E₁-E₄ correspond to CO₂-CO₂ packing with densities of 1.05, 0.23, 0.018, and 0.003 g/cm³, respectively (we note that F₂ and F₃ have negligible contributions and are not shown; see **Figure S127** for further information). **c**, Illustrations of the spatial distributions of sub-contexts F_{1,1}, F_{1,2}, F_{1,3}, F_{1,4}.

2.4. Molecular packing in the active layer of the PM6:Y6 organic solar cell

Finally, we illustrate the use of the HiDiscover protocol to investigate the molecular packing of Y6, a state-of-the-art non-fullerene small-molecule acceptor that has been extensively utilized in binary or ternary active layers in organic photovoltaics.⁶³ However, the microscopic states of Y6 during device operation remain poorly understood, complicated further by its complex molecular structure that makes the elucidation of the electron transport pathways challenging.^{64,65} In the active layers of the bulk heterojunction organic solar cells, which correspond to blends of electron-donor and electron-acceptor components, various domains with different Y6 ratios exist. Hole-electron pairs are often generated at the donor-acceptor interfaces or in mixed phases; these holes and electrons need to transport to their respective collecting electrodes efficiently for the solar cell to achieve high performance. Currently, the Y6 molecular configurations and the electron-transport pathways they form in different regions of the solar-cell active layers remain poorly understood. Here, we examine blends of PM6 (a widely used donor polymer) with Y6 at varying donor-acceptor ratios (**Figure 6a**) to probe the electron transport pathways from a donor-rich domain to a mixed domain and then to an acceptor domain, corresponding to the direction of electron transport to the electrode. We note that relying on a conventional molecular simulation protocol is expected to miss the quantitative connections between the various regions in the blend. Here, the HiDiscover protocol enables us to establish quantitatively these connections among domains across space, which allows for a better characterization of the electron-transport pathways over long distances and advances our understanding of global morphological features in organic photovoltaic active layers. Our analysis focuses on the short-range Y6-Y6 contacts (task

ζ_G), which is an indicator for the electron transport pathways (as well as for potential charge generation within acceptor domains).⁴⁸

Figure 6. Illustration of application of the HiDiscover protocol to study the electron transport pathways in different regions of the PM6:Y6 bulk heterojunction. **a**, Molecular models as a function of Y6 ratio (the Y6 side chains are not shown and the PM6 chains are displayed in grey). **b**, Short-range Y6-Y6 contacts identified in the crystalline region and the ones that appear in the amorphous domains and are not present in the crystalline domains. **c**, Ratios of short-range Y6-Y6 contacts in various of in the PM6:Y6 blend.

Notably, in both the mixed and acceptor domains, around 50% of short-range Y6-Y6 contacts resemble those observed in the Y6 single crystals (**Figure 6bc**). At PM6:Y6 ratios from 4:1 to 1:2 (wt), the portion of crystalline-like molecular contacts is even higher than in the pure Y6 phase (**Figure 6c**). Specifically, tail-to-tail ($G_{1,1}$ and $G_{1,2}$) and tilted ($G_{1,4}$) Y6-Y6 interactions account for approximately 50% of these short-range contacts. Compared to the disordered Y6-Y6 configurations found only within the amorphous

packing (G_2 , see **Figure 6b**), these crystalline-like configurations exhibit well-overlapped π -conjugation areas, which can lead to efficient electron transport pathways throughout the mixed and acceptor domains, thereby contributing to the high performance of the Y6 acceptor. Interestingly, the configuration with the highest overlap, $G_{1.3}$, is less prevalent than $G_{1.1}$, $G_{1.2}$, and $G_{1.4}$, particularly in donor-rich domains. This points to opportunities for improving electron transport efficiencies through further molecular engineering. These findings underscore the value of probing the detailed microscopic features in complex organic solar-cell active layers and pave the way for discovering optimal molecular configurations in more complex ternary systems in which an additional donor or acceptor component is present and which lead to the highest reported power conversion efficiencies, now over 20% for organic photovoltaics.^{65,66}

3. Discussion

Machine learning approaches have considerably impacted materials research in recent years. Notably, machine learning force fields have been developed that allow for simulating larger systems with accuracies approaching ab initio methods at reduced computational cost.^{67,68} These methods have greatly expanded materials energy and environmental research, which often deals with complex systems that call for the modeling of large systems. However, an equally important task lies in apprehending the results from the modeling of the complex systems. While ML-based approaches have been greatly advanced to analyze molecular simulation data,^{43,69,70,45,27,71} general methods to identify meaningful atomic/molecular arrangements within complex, multi-component materials frequently encountered in energy and environmental research have not been explored. This work establishes effective methodologies to identify meaningful patterns of atomic and

molecular arrangements in these advanced materials, addressing a major challenge in their microscopic mechanistic studies. Specifically, we have developed a novel research protocol named HiDiscover and enabled the systematic extraction of microscopic configurational information in intricate molecular systems, which is crucial to mechanistic understandings of materials to inform their design, but challenging to probe by conventional analysis. This approach is particularly suitable for investigating multi-component systems with various interfaces frequently encountered in advanced materials, energy and environmental applications, as we demonstrated in the cases of Li-ion transport in a covalent organic framework, gas adsorption in a metal-organic framework, and molecular packing in organic photovoltaics.

In the context of Li-ion transport within 2D COFs, prior investigations have largely been empirical, constrained by a lack of methods capable of probing detailed microscopic conformations. This limitation has impeded progress in optimizing solid electrolytes. Our approach addresses this challenge, revealing detailed features of Li-ion transport in COF-PEO-3. These offer insights crucial for guiding the future design of 2D COFs aimed at enhancing ionic conductivity. Regarding gas adsorption in MOFs, our innovative approach facilitated the characterization of CO₂ transport pathways within MOF-5, which represent important features for designing materials for molecular separation applications yet overlooked by previous simulations. Finally, using the HiDiscover protocol, we have quantitatively characterized the molecular packing of the Y6 acceptor across various domains in organic solar cells, which is a critical determinant of electron transport efficiency across long distances but remains understudied due to its inherent complexity.

These findings highlight the potential of our method to advance material research through the integration of machine intelligence.

By applying the HiDiscover protocol, we can derive quantitative microscopic features from exhaustive machine-learned interpretations in a standardized way. Compared to the conventional practice that heavily relies on researchers directly analyzing original molecular simulation data (often in a customized way influenced by the researcher's experience), the HiDiscover protocol also mitigates the issue of incomplete observations made by individuals. Consequently, it minimizes bias and enhances the reliability of the resulting mechanisms.

We note that the HiDiscover protocol can be used with common molecular simulation techniques, making it applicable to the study of a wide range of materials and shifting computational materials research from qualitative and isolated observations towards more quantitative and comprehensive descriptions. **At this stage, we anticipate no impediments to the application of the HiDiscover protocol to other complex, multi-component materials beyond those discussed here.** In combination with already well-established molecular simulation tools, it can significantly contribute to advancing our understanding of microscopic mechanisms in materials at the atomic and molecular levels, thereby accelerating the design and optimization of advanced materials.

As discussed above, the objective of HiDiscover is to identify meaningful atomic and molecular arrangements within complex, multi-component materials. To the best of our knowledge, no general methodology, whether ML-based or otherwise, currently exists to address this complexity. It is useful here to note some of the design considerations as well as limitations of the HiDiscover protocol for its application and future development:

(1) Since our focus is on identifying atomic and molecular arrangements (static features), the temporal correlation in the MD trajectory is not treated in the model training. Instead, the simulated frames are treated as uncorrelated samplings of the systems around their equilibrium states. The temporal correlation in the dataset is generally weak and has very little impact on the results (see **Section 4.4** in the **SI**). Although the temporal correlation in the data is not learned, mechanisms pertaining to temporal evolution can still be inferred by mapping predicted labels onto MD trajectories for analysis, as exemplified in our study on Li-ion transport.

(2) To most effectively use the HiDiscover protocol, a reasonable prior knowledge of the studied material is required. It is worth stressing that this prerequisite is not unique to HiDiscover but is implicitly present in traditional mechanistic studies of materials through molecular modeling as well, albeit often unacknowledged. Thus, we emphasize that insights gained from existing research, preliminary simulations, and even human observations can facilitate the use of the HiDiscover protocol as well.

(3) It is important to note that the HiDiscover protocol does not automatically identify the required tasks and relies on user guidance instead, similar to the “interest points” of a researcher in a conventional MD study. In a conventional MD study, the researcher may observe the data and adjust the “interest points” as the study is performed. Similarly, the “tasks” in a HiDiscover protocol may also be tuned if the researcher gathers new information (*e.g.*, from new research, preliminary observations, and analysis). We emphasize that, explicitly defining the “tasks” instead of implicitly using the “interest points” of the researchers is expected to improve the reproducibility of material mechanistic studies. A fact in materials modeling is that, while the calculated properties

(*e.g.*, mobilities, diffusivities) are often reproducible across different publications, the proposed mechanisms for the same material are often highly researcher-dependent as researchers typically focus on different aspects based on their experience (a fact that may not be documented along with the discovered mechanisms). In a HiDiscover protocol, the tasks are to be explicitly provided along with the discovered mechanisms, thus promoting the reproducibility of materials mechanistic studies.

(4) Reference molecular models are also frequently involved in conventional molecular dynamics studies. In the HiDiscover protocol, we take advantage of a set of reference systems for machine learning, allowing us to derive quantitative features of the atomic and molecular arrangements. Given the diversity of multi-component materials in energy and environmental research and the broad range of the mechanisms of interest, the methods to design the reference models are also expected to be versatile. For studies using molecular modeling, determining the reference systems is often obvious to the researcher since the objective is well-defined and a good prior understanding has been (or will be) obtained. Here, we have discussed four different types of materials and carried out calculations for three of them, demonstrating the feasibility of designing reference systems in the context of mechanistic studies of specific materials. These examples encompass solutions, porous composite materials, and organic heterojunctions at varying concentrations and degrees of mixing, which collectively serve as a reference for applying the HiDiscover protocol to other types of multi-component materials. A rule of thumb in the design of reference systems is to use simpler molecular models comprised of partial components of the target multi-component material, which are generally easier to construct and share similarities to

the studied material. To aid this process, we have provided reference system suggestions for general multi-component materials (see **Section 6** and **Table S4** in the **SI**).

(5) In the HiDiscover protocol, the reference systems do not exclude the target system. For example, if our focus is on the impact of concentration on the molecular arrangements, a series of target systems of varying concentrations would be built even in a conventional MD study, which naturally becomes the complete reference systems in the HiDiscover protocol (as in task ζ_F). In other cases, it is still suggested to set the target system (or representative ones from the target systems) in the reference systems to include all pertinent atomic/molecular arrangements as done in this study, except when the researcher is confident that all pertinent features in all the target systems have been included in the reference systems. An example would be minor variations of the target systems (*e.g.*, small concentration or temperature changes), which are unlikely to introduce significant new atomic/molecular arrangements and thus only a single representative target system may be included in the reference systems.

(6) If we have included the target system in the reference systems and only a small portion of predicted labels correspond to the other reference molecular systems, we will know that the selected reference systems do not contain the majority of the atomic and molecular arrangements present in the target multi-component material. This result tells us how different the target system is from the remaining reference systems, which in itself can also be useful information depending on the researcher's goal. In this case, all features of atomic and molecular arrangements in the multi-component material are still captured, although they cannot be linked to the other reference systems for easy human interpretation. Optimizing the list of reference molecular models may be desirable for the researcher,

similar to the process of designing molecular models iteratively often done in a conventional MD study.

(7) The HiDiscover protocol may introduce new molecular models when the target systems being studied do not cover all reference systems. To sample sufficient atomic/molecular arrangements, parallel MD simulations with different initial configurations are recommended for the reference systems. Long MD simulations are also preferable. These are expected to increase the computational cost compared to only simulating the target systems in a conventional MD approach. In this study, we have tried to run multiple long MD simulations for the reference systems within our computational resources. Importantly, depending on the studied problem, using the HiDiscover protocol only leads to 0-150% increase in the computational time compared to studying the same problems with conventional MD simulations (that cannot yield the same quantitative insights as in the HiDiscover protocol). Detailed timing information can be found in **Section 4** of the **SI**. In general, the increased computational time when using the HiDiscover protocol compared to conventional MD study is related to the cost associated with new molecular models. Very interestingly, we have also performed tests on the dataset size for the reference systems and found that using just the first 1% of the MD production runs can also achieve robust results (see also **Section 4** of the **SI**).

(8) To derive reliable material mechanisms, long MD simulations are usually needed for the target systems, for reasons similar to that discussed above. In this respect, using HiDiscover or a conventional protocol is not different. As such, the researchers still need to pay attention to the simulations on the target systems when using HiDiscover to investigate materials mechanisms.

4. Methods

4.1. Design of the contexts for Li-ion transporting in COF-PEO-3

Within the composite system of COF-PEO-3 electrolyte, multiple phases coexist, leading to the formation of various interfaces. Our previous investigation indicated that salt aggregates into clusters within the 2D COF pores.⁴⁶ These clusters can be in contact with the COF framework or isolated from other components. Building upon this understanding, we differentiate in task ζ_A between $\text{Li}^+\text{-ClO}_4^-$ configurations found in crystalline LiClO_4 (A_1), amorphous LiClO_4 (A_2), at the surfaces of LiClO_4 clusters (A_3), those emerging when LiClO_4 is blended with $(\text{PEO})_3$ (A_4), those near the COF framework (A_5), and the additional configurations arising in the target system (A_6).

As a prior study has suggested different ion transport pathways at varying distances from the 2D COF framework,⁴⁷ we distinguish in tasks ζ_B and ζ_C between ions near the 2D COF framework (B_1 and C_1), those within the pores (B_2 and C_2), and new configurations emerging in the target system (B_3 and C_3). Regarding task ζ_D , we differentiate between the configurations of the $\text{Li}^+\text{-PEO}$ chain in bulk $(\text{PEO})_3$ (D_1) and the new ones emerging in COF-PEO-3 (D_2), recognizing that the framework imposes restrictions on the PEO chains and potentially influences their arrangements around the Li-ion.

4.2. Molecular dynamics simulations

COF-PEO-3. We employed the all-atom optimized potentials for liquid simulations (OPLS-AA) force field⁷², which has been widely applied in 2D COF studies for its efficiency and accuracy.⁷³⁻⁷⁷ MD model 1 corresponds to an $8\times 6\times 5$ supercell of LiClO_4 . MD model 2 contains 1000 LiClO_4 molecules randomly placed in a box with an initial size

of 4.5 nm × 4.5 nm × 4.5 nm. MD model 3 is a LiClO₄ cluster, built from the final molecular structure of MD model 2, with the new box size set to 9 nm × 9 nm × 9 nm. MD models 4-6 contain 60, 30, and 240 LiClO₄ molecules (corresponding to Li⁺ ratios of 0.46wt%, 0.24 wt%, and 1.5 wt%, respectively) mixed in 2×2×20 supercells of COF-42, respectively. MD model 7 contains 60 LiClO₄ and 480 (PEO)₃ in a box with an initial size of 6 nm × 6 nm × 6 nm. In MD model 8, 360 LiClO₄ were initially randomly blended in a 2×2×20 supercell of COF-PEO-3. The ratio of Li⁺ is 1.4 wt%, close to that used experimentally⁴³. Four parallel structures were constructed for MD models 2-8. Details can be found in **Section 2** of the **SI**.

MOF-5. We turned to the Universal force field, which has been widely employed in studying gas adsorption in MOFs.^{57,58,78} In MD models 9-12, 1000 CO₂ molecules were initially randomly placed in a box; pressures of 1, 10, 100, and 1000 bars were applied, respectively. In MD models 13-16, 25, 99, 493, and 985 CO₂ were initially randomly put in a 2×2×2 supercell of MOF-5; these correspond to concentrations of 0.5, 2, 10, and 30 mmol/g, respectively. Since these systems are relatively well mixed, we use a single initial structure for each MD model. Details can be found in **Section 2** of the **SI**.

PM6:Y6 heterojunction. We used the OPLS-AA force field, which has been extensively considered for the description of organic solar-cell active layers.^{72,79,80} MD models of crystalline Y6, amorphous Y6, and PM6:Y6 with different weight ratios were constructed. In MD model 17, a 2×1×2 supercell of Y6 was constructed based on its crystal structure.⁴⁸ This model is used to learn the short-range contacts in Y6 crystals. In MD model 18, 100 Y6 molecules were initially randomly placed in a box; this model represents the pure acceptor phase. In MD models 19-22, 100 Y6 and 3, 6, 12, or 24 PM6 chains consisting of

20 repeat units were initially randomly placed in a box; these represent regions going from an acceptor-rich domain to a donor-rich domain. Sixteen parallel structures were constructed for each MD model except for Model 17. Details can be found in **Section 2** of the **SI**.

Simulation details. Unless otherwise mentioned, each MD model underwent an initial energy minimization (steepest descend algorithm), followed by equilibrium and production runs. The short-range electrostatic and van der Waals cutoffs were set to 1.4 nm. The smooth particle-mesh Ewald (PME) method was used for long-range electrostatic interactions.⁸¹ A velocity rescaling scheme was considered for thermostat⁸² and Berendsen, for barostat⁸³. The time step was set to 1 fs. During the production run, structures were output every 10 ps. All molecular dynamics simulations were performed using the GROMACS package.⁸⁴

Notes on the molecular models. To ensure the robustness of the incremental learning model, it is necessary that the molecular dynamics data used in the preceding training step encompass the relevant ionic arrangements for the current class-incremental training. For example, MD model 1 should generate a substantial amount of ionic arrangements to include those belonging to context A₁ as observed in MD models 2, 3, 4, 7, and 8. This point should be given special attention in the case of Li-ion transport in COF-PEO-3 that exhibits inhomogeneous mixing. The dataset also needs to be sufficiently large to capture all significant ionic arrangements relevant to its context in the subsequent training as well. This can be achieved by conducting multiple long MD simulations on well-equilibrated systems with sufficient sampling. The sizes of the datasets range from $\sim 10^5$ - 10^7 , as listed in **Table S2**. Tests on the size of the dataset can be found in **Section 4** of the **SI**.

Notes on the relevance of the contexts. We acknowledge that the depiction of microscopic mechanisms inherently involves a certain degree of subjectivity. When analyzing the atomic/molecular arrangements in molecular simulation trajectories, human interpretations heavily rely on intuition and experience. In previous machine learning models designed to classify atomic arrangements, these interpretations corresponded to the data labels, often categorized as “solids”, “solid-liquid interface”, “fcc-type crystal”, etc. Within our incremental learning approach, the defined context set introduces language elements through which machine-learned features are expressed. Ideally, these defined contexts should align with human comprehension, making them easily understandable. In practical terms, the context set can relate to the attributes of simpler molecular systems, establishing connections to the complex problem at hand. This approach provides a flexible method for introducing high-resolution labels and allows for detailed differentiation of intricate components and interfaces in the target system; this facilitates the extraction of deep insights from the simulation data and a more comprehensive description of the microscopic mechanisms. The primary limitation in defining the contexts lies in the feasibility of constructing the necessary dataset from molecular modeling.

4.3. Data processing

COF-PEO-3. To process the raw data obtained from molecular dynamics simulations, we extract the local environment of each Li^+ , represented by the coordinates of selected groups of atoms surrounding it. Considering that neighboring species often have identical atoms (*e.g.*, the four oxygen atoms in ClO_4^-), as each coordinating with the ion would lead to similar configurations, we define equivalent atom groups to encompass multiple atoms in a molecule. The closest of the atoms to the ion is used to calculate the relative position of

an equivalent atom group. This also reduces the record length for easier model training. Additionally, symmetry and sort operations were performed. These relative positions are subsequently converted into a Coulomb matrix format to generate the dataset. The dataset is then divided into training, validation, and test sets at an 8:1:1 ratio. Further information regarding the data processing procedure can be found in **Section 3** of the **SI**.

MOF-5 and PM6:Y6 heterojunction. The data processing procedure is similar to that used for COF-PEO-3. For MOF-5, we focus on the configurations among CO₂ molecules and the arrangement of CO₂ with respect to the metal oxide core and phenyl ring linker. For PM6:Y6, we focus on the short-range Y6-Y6 contacts. Detailed information can be found in **Section 3** of the **SI**.

4.4. Model training

COF-PEO-3. We first train the class-incremental model within the inner layer of the hierarchical incremental learning framework. As shown in **Figure 3a**, we initially applied unsupervised learning on the training set from MD model 1, employing k-means clustering, which offers a high level of interpretability. This resulted in classification model 1 with centroids corresponding to Li⁺-ClO₄⁻ configurations found in LiClO₄ crystals. Subsequently, the training set derived from MD model 2 was employed for class-incremental learning using a modified k-means algorithm, leading to classification model 2. This particular model possesses the capability to distinguish the context of Li⁺-ClO₄⁻ between crystalline and amorphous states. Then, four additional iterations of class-incremental learning were sequentially performed, using the training sets obtained from MD models 3, 7, 4, and 8. This ultimately leads to classification model 6 for task $\zeta_A (x_A \rightarrow A \times y_A)$, gaining the new ability to identify Li⁺-ClO₄⁻ configurations on cluster surfaces,

within interfacial regions in the PEO-chain environment, or next to the 2D COF framework, as well as those emerging in COF-PEO-3.

For Li^+ -COF configurations, the datasets corresponding to MD models 4, 5, and 8 are used. Unsupervised learning was employed to generate classification models 7 and 10 using the training sets from MD model 4. Subsequently, the training sets obtained from MD models 5 and 8 were utilized for class-incremental learning, eventually leading to classification models 9 and 12 for tasks $\zeta_B (x_B \rightarrow B \times y_B)$ and $\zeta_C (x_C \rightarrow C \times y_C)$, respectively. Regarding Li^+ -PEO configurations (tasks ζ_D), a similar process of incremental learning is conducted using datasets derived from MD models 7 and 8. Consequently, classification model 14 is obtained ($x_D \rightarrow D \times y_D$), effectively characterizing the configurations related to the Li^+ -PEO side-chain interactions in the target system. Conducting sequential training of all the inner class-incremental learning models corresponds to task-incremental learning in the outer layer.

Details of the training processes and the determination of the k values can be found in **Section 4.1** of the **SI**. Accuracies of $> 99\%$ are achieved in our incremental learning model.

MOF-5 and PM6:Y6 heterojunction. The model training procedure is similar to that for COF-PEO-3. Accuracies of $> 99\%$ are again achieved in our incremental learning model.

Details can be found in **Sections 4.2** and **4.3** of the **SI**.

Notes on the sequence of datasets. The order of the contexts Ω fed into the model is a factor in determining the labels. There are four possible scenarios for two successive instances Ω_i and Ω_j : (i) When there is no overlap in context between Ω_i and Ω_j , the situation corresponds to the conventional case of acquiring labeled data. In this instance, switching

the order of Ω_i and Ω_j will have no impact. (ii) If Ω_i is a subset of Ω_j , the smaller subset should be considered first, otherwise it will be represented by the larger dataset and thus will not contribute to determining the labels in the target system. (iii) When Ω_i and Ω_j have overlapping contexts but also contain parts exclusive to each other, the overlapped context will be incorporated into the label associated with the first dataset. (iv) Ω_i and Ω_j may also have identical contexts, in which case only one is needed. In general, it is desired to gradually increase the complexity of the datasets. In practice, to determine Ω and their order, researchers can rely on their background knowledge of the complexity of the molecular systems and verify it during model training. For instance, in case (ii), reversing the order would lead to the second label having minimal data coverage, thereby providing little assistance in distinguishing the atomic and molecular arrangements; in case (iv), adding Ω_i and Ω_j together would lead to severely reduced accuracy of the model. Detailed considerations for the determination of the dataset order in this work can be found in **Section 4** of the SI.

4.5. Analysis of MD data with machine-generated interpretations

By applying the trained model to the data obtained from an MD trajectory of the target system, one can obtain a temporal sequence of labels that depict the evolution of microscopic states. Specifically, for COF-PEO-3, the final classification model was employed on the dataset derived from the MD simulations of COF-PEO-3 filled with LiClO_4 , generating the labels for each Li^+ with various components at different frames. For MOF-5, the final classification model was employed on the MD systems with different CO_2 loadings. For the PM6:Y6 heterojunction, the final classification model was applied to the MD systems with different PM6:Y6 ratios. We then perform analysis on these

machine-learned labels, reducing the reliance on directly analyzing the raw MD data as done in a conventional study routine. In particular, we calculated the ratio of each label in the target system and their correlation coefficient. In the time sequence analysis, to suppress the impact of errors from the machine-learning model, we use a criterion of 3 or more successive occurrences (corresponding to a 30-ps time window) of the same (or different) states to determine the start (or end) of a state.

Supplementary Materials

A pdf file containing the mathematical description of HiDiscover, discussions on the impact of temporal correlation in the dataset, summary of the computational cost, additional information on the MD simulations, data processing, model training, analyses, and suggestions for designing the reference molecular systems.

Data availability

Samples of the datasets generated in this study have been deposited at <https://zenodo.org/record/13292810>.

Code availability

The code and tutorial for model training are available in the attachment.

Acknowledgments

The work at Shanghai University was supported by the National Natural Science Foundation of China (grant numbers: 22473072 and 22103053) and the Shanghai Technical Service Center of Science and Engineering Computing at Shanghai University. The work at Washington University was supported by the National Science Foundation Grant DMS-2418979. The work at Jilin University was financially supported by the National Key R&D Program of China (grant number: 2023YFB3003001) and "Xiaomi Young Scholar" Project. The work at Arizona was funded by the Office of Naval Research, Award No. N00014-24-1-2114; the authors are also grateful to the University of Arizona Institute of Energy Solutions and Office for Research, Innovation, and Impact for support via the Arizona Technology and Research Initiative Fund; the computational work was supported in part by a grant of computer time from the DoD High Performance Modernization Program.

References

1. Sumaria, V., Nguyen, L., Tao, F. F. & Sautet, P. Atomic-Scale Mechanism of Platinum Catalyst Restructuring under a Pressure of Reactant Gas. *J. Am. Chem. Soc.* **145**, 392–401 (2023).

2. Wang, C. *et al.* The molecular mechanism of constructive remodeling of a mechanically-loaded polymer. *Nat. Commun.* **13**, 3154 (2022).
3. Luo, T., Mohan, K., Iglesias, P. A. & Robinson, D. N. Molecular mechanisms of cellular mechanosensing. *Nat. Mater.* **12**, 1064–1071 (2013).
4. Shah, N. H. & Kuriyan, J. Understanding molecular mechanisms in cell signaling through natural and artificial sequence variation. *Nat. Struct. Mol. Biol.* **26**, 25–34 (2019).
5. Arosio, P. *et al.* Kinetic analysis reveals the diversity of microscopic mechanisms through which molecular chaperones suppress amyloid formation. *Nat. Commun.* **7**, 10948 (2016).
6. Hu, L., Huang, B. & Liu, F. Atomistic Mechanism Underlying the Si(111)-(7×7) Surface Reconstruction Revealed by Artificial Neural-Network Potential. *Phys. Rev. Lett.* **126**, 176101 (2021).
7. Hollingsworth, S. A. & Dror, R. O. Molecular Dynamics Simulation for All. *Neuron* **99**, 1129–1143 (2018).
8. Badocha, M. *et al.* Molecular mechanism and energetics of coupling between substrate binding and product release in the F1-ATPase catalytic cycle. *Proc. Natl. Acad. Sci.* **120**, e2215650120 (2023).
9. Yoon, B. & Voth, G. A. Elucidating the Molecular Mechanism of CO₂ Capture by Amino Acid Ionic Liquids. *J. Am. Chem. Soc.* **145**, 15663–15667 (2023).
10. Fu, F. *et al.* Unraveling the Atomic-scale Mechanism of Phase Transformations and Structural Evolutions during (de)Lithiation in Si Anodes. *Adv. Funct. Mater.* **33**, 2303936 (2023).
11. Xiouras, C. *et al.* Applications of Artificial Intelligence and Machine Learning Algorithms to Crystallization. *Chem. Rev.* **122**, 13006–13042 (2022).
12. Zhou, B. *et al.* Light-driven synthesis of C₂H₆ from CO₂ and H₂O on a bimetallic AuIr composite supported on InGaN nanowires. *Nat. Catal.* 1–9 (2023) doi:10.1038/s41929-023-01023-1.
13. Wang, X. *et al.* Mechanical nonreciprocity in a uniform composite material. *Science* **380**, 192–198 (2023).
14. Hurtig, F. *et al.* The patterned assembly and stepwise Vps4-mediated disassembly of composite ESCRT-III polymers drives archaeal cell division. *Sci. Adv.* **9**, eade5224 (2023).
15. Xue, T. *et al.* A customized MOF-polymer composite for rapid gold extraction from water matrices. *Sci. Adv.* **9**, eadg4923 (2023).
16. Xu, Y., Ma, Y.-B., Gu, F., Yang, S.-S. & Tian, C.-S. Structure evolution at the gate-tunable suspended graphene–water interface. *Nature* **621**, 506–510 (2023).
17. Chin, X. Y. *et al.* Interface passivation for 31.25%-efficient perovskite/silicon tandem solar cells. *Science* **381**, 59–63 (2023).
18. Liu, X. *et al.* Nonlinear optical phonon spectroscopy revealing polaronic signatures of the LaAlO₃/SrTiO₃ interface. *Sci. Adv.* **9**, eadg7037 (2023).
19. Mariotti, S. *et al.* Interface engineering for high-performance, triple-halide perovskite–silicon tandem solar cells. *Science* **381**, 63–69 (2023).
20. Jeong, K. *et al.* Solvent-Free, Single Lithium-Ion Conducting Covalent Organic Frameworks. *J. Am. Chem. Soc.* **141**, 5880–5885 (2019).

21. Geng, K. *et al.* Covalent Organic Frameworks: Design, Synthesis, and Functions. *Chem. Rev.* **120**, 8814–8933 (2020).
22. Li, X. *et al.* Solution-Processable Covalent Organic Framework Electrolytes for All-Solid-State Li–Organic Batteries. *ACS Energy Lett.* **5**, 3498–3506 (2020).
23. Kang, T. W. *et al.* An Ion-Channel-Restructured Zwitterionic Covalent Organic Framework Solid Electrolyte for All-Solid-State Lithium-Metal Batteries. *Adv. Mater.* **35**, 2301308 (2023).
24. Zhang, H. *et al.* Charge and mass transport mechanisms in two-dimensional covalent organic frameworks (2D COFs) for electrochemical energy storage devices. *Energy Environ. Sci.* **16**, 889–951 (2023).
25. Li, C. *et al.* Anthraquinone-Based Silicate Covalent Organic Frameworks as Solid Electrolyte Interphase for High-Performance Lithium–Metal Batteries. *J. Am. Chem. Soc.* **145**, 24603–24614 (2023).
26. Venable, R. M., Krämer, A. & Pastor, R. W. Molecular Dynamics Simulations of Membrane Permeability. *Chem. Rev.* **119**, 5954–5997 (2019).
27. Yao, N., Chen, X., Fu, Z.-H. & Zhang, Q. Applying Classical, Ab Initio, and Machine-Learning Molecular Dynamics Simulations to the Liquid Electrolyte for Rechargeable Batteries. *Chem. Rev.* **122**, 10970–11021 (2022).
28. Bedrov, D. *et al.* Molecular Dynamics Simulations of Ionic Liquids and Electrolytes Using Polarizable Force Fields. *Chem. Rev.* **119**, 7940–7995 (2019).
29. Honeycutt, J. Dana. & Andersen, H. C. Molecular dynamics study of melting and freezing of small Lennard-Jones clusters. *J. Phys. Chem.* **91**, 4950–4963 (1987).
30. Kelchner, C. L., Plimpton, S. J. & Hamilton, J. C. Dislocation nucleation and defect structure during surface indentation. *Phys. Rev. B* **58**, 11085–11088 (1998).
31. Tsuzuki, H., Branicio, P. S. & Rino, J. P. Structural characterization of deformed crystals by analysis of common atomic neighborhood. *Comput. Phys. Commun.* **177**, 518–523 (2007).
32. Ackland, G. J. & Jones, A. P. Applications of local crystal structure measures in experiment and simulation. *Phys. Rev. B* **73**, 054104 (2006).
33. Larsen, P. M., Schmidt, S. & Schiøtz, J. Robust structural identification via polyhedral template matching. *Model. Simul. Mater. Sci. Eng.* **24**, 055007 (2016).
34. Radhi, A., Iacobellis, V. & Behdinin, K. A Cumulative Approach to Crystalline Structure Characterization in Atomistic Simulations. *J. Phys. Chem. C* **122**, 13156–13165 (2018).
35. Fukuya, T. & Shibuta, Y. Machine learning approach to automated analysis of atomic configuration of molecular dynamics simulation. *Comput. Mater. Sci.* **184**, 109880 (2020).
36. Allera, A., Goryaeva, A. M., Lafourcade, P., Maillet, J.-B. & Marinica, M.-C. Neighbors Map: An efficient atomic descriptor for structural analysis. *Comput. Mater. Sci.* **231**, 112535 (2024).
37. Ryan, K., Lengyel, J. & Shatruk, M. Crystal Structure Prediction via Deep Learning. *J. Am. Chem. Soc.* **140**, 10158–10168 (2018).
38. Chung, H. W., Freitas, R., Cheon, G. & Reed, E. J. Data-centric framework for crystal structure identification in atomistic simulations using machine learning. *Phys. Rev. Mater.* **6**, 043801 (2022).

39. Freitas, R. & Reed, E. J. Uncovering the effects of interface-induced ordering of liquid on crystal growth using machine learning. *Nat. Commun.* **11**, 3260 (2020).
40. Du, T. *et al.* Predicting Fracture Propensity in Amorphous Alumina from Its Static Structure Using Machine Learning. *ACS Nano* **15**, 17705–17716 (2021).
41. Liu, H., Smedskjaer, M. M. & Bauchy, M. Deciphering a structural signature of glass dynamics by machine learning. *Phys. Rev. B* **106**, 214206 (2022).
42. Deng, Y., Wang, Y., Xu, K. & Wang, Y. Lightweight Extendable Stacking Framework for Structure Classification in Atomistic Simulations. *J. Chem. Theory Comput.* **19**, 8332–8339 (2023).
43. Noé, F., Tkatchenko, A., Müller, K.-R. & Clementi, C. Machine Learning for Molecular Simulation. *Annu. Rev. Phys. Chem.* **71**, 361–390 (2020).
44. Glielmo, A. *et al.* Unsupervised Learning Methods for Molecular Simulation Data. *Chem. Rev.* **121**, 9722–9758 (2021).
45. Kaptan, S. & Vattulainen, I. Machine learning in the analysis of biomolecular simulations. *Adv. Phys. X* **7**, 2006080 (2022).
46. van de Ven, G. M., Tuytelaars, T. & Tolias, A. S. Three types of incremental learning. *Nat. Mach. Intell.* **4**, 1185–1197 (2022).
47. Furukawa, H., Cordova, K. E., O’Keeffe, M. & Yaghi, O. M. The Chemistry and Applications of Metal-Organic Frameworks. *Science* vol. 341 1230444 (2013).
48. Zhu, W. *et al.* Crystallography, Morphology, Electronic Structure, and Transport in Non-Fullerene/Non-Indacenodithienothiophene Polymer:Y6 Solar Cells. *J. Am. Chem. Soc.* **142**, 14532–14547 (2020).
49. Shao, J., Tanner, S. W., Thompson, N. & Cheatham, T. E. Clustering Molecular Dynamics Trajectories: 1. Characterizing the Performance of Different Clustering Algorithms. *J. Chem. Theory Comput.* **3**, 2312–2334 (2007).
50. Zhang, G. *et al.* Accumulation of Glassy Poly(ethylene oxide) Anchored in a Covalent Organic Framework as a Solid-State Li⁺ Electrolyte. *J. Am. Chem. Soc.* **141**, 1227–1234 (2019).
51. Xu, Q., Tao, S., Jiang, Q. & Jiang, D. Ion Conduction in Polyelectrolyte Covalent Organic Frameworks. *J. Am. Chem. Soc.* **140**, 7429–7432 (2018).
52. Guo, Z. *et al.* Fast Ion Transport Pathway Provided by Polyethylene Glycol Confined in Covalent Organic Frameworks. *J. Am. Chem. Soc.* **141**, 1923–1927 (2019).
53. Skoulidas, A. I. Molecular Dynamics Simulations of Gas Diffusion in Metal–Organic Frameworks: Argon in CuBTC. *J. Am. Chem. Soc.* **126**, 1356–1357 (2004).
54. Zhao, Z., Li, Z. & Lin, Y. S. Adsorption and Diffusion of Carbon Dioxide on Metal–Organic Framework (MOF-5). *Ind. Eng. Chem. Res.* **48**, 10015–10020 (2009).
55. Saha, D., Bao, Z., Jia, F. & Deng, S. Adsorption of CO₂, CH₄, N₂O, and N₂ on MOF-5, MOF-177, and Zeolite 5A. *Environ. Sci. Technol.* **44**, 1820–1826 (2010).
56. Qian, Q. *et al.* MOF-Based Membranes for Gas Separations. *Chem. Rev.* **120**, 8161–8266 (2020).
57. Skoulidas, A. I. & Sholl, D. S. Self-Diffusion and Transport Diffusion of Light Gases in Metal-Organic Framework Materials Assessed Using Molecular Dynamics Simulations. *J. Phys. Chem. B* **109**, 15760–15768 (2005).
58. Atci, E., Erucar, I. & Keskin, S. Adsorption and Transport of CH₄, CO₂, H₂ Mixtures in a Bio-MOF Material from Molecular Simulations. *J. Phys. Chem. C* **115**, 6833–6840 (2011).

59. Rowsell, J. L. C., Spencer, E. C., Eckert, J., Howard, J. A. K. & Yaghi, O. M. Gas Adsorption Sites in a Large-Pore Metal-Organic Framework. *Science* vol. 309 1350–1354 (2005).
60. García-Pérez, E., Serra-Crespo, P., Hamad, S., Kapteijn, F. & Gascon, J. Molecular simulation of gas adsorption and diffusion in a breathing MOF using a rigid force field. *Phys. Chem. Chem. Phys.* **16**, 16060–16066 (2014).
61. Pillai, R. S., Pinto, M. L., Pires, J., Jorge, M. & Gomes, J. R. B. Understanding Gas Adsorption Selectivity in IRMOF-8 Using Molecular Simulation. *ACS Appl. Mater. Interfaces* **7**, 624–637 (2015).
62. Listyarini, R. V., Gamper, J. & Hofer, T. S. Storage and Diffusion of Carbon Dioxide in the Metal Organic Framework MOF-5—A Semi-empirical Molecular Dynamics Study. *J. Phys. Chem. B* **127**, 9378–9389 (2023).
63. Shoaee, S. *et al.* What We have Learnt from PM6:Y6. *Adv. Mater.* **36**, 2302005 (2024).
64. Li, C. *et al.* Non-fullerene acceptors with branched side chains and improved molecular packing to exceed 18% efficiency in organic solar cells. *Nat. Energy* **6**, 605–613 (2021).
65. He, C. *et al.* Asymmetric electron acceptor enables highly luminescent organic solar cells with certified efficiency over 18%. *Nat. Commun.* **13**, 2598 (2022).
66. Dolan, A. *et al.* Enhanced Photocatalytic and Photovoltaic Performance Arising from Unconventionally Low Donor–Y6 Ratios. *Adv. Mater.* **36**, 2309672 (2024).
67. Unke, O. T. *et al.* Machine Learning Force Fields. *Chem. Rev.* **121**, 10142–10186 (2021).
68. Zhang, L., Han, J., Wang, H., Car, R. & E, W. Deep Potential Molecular Dynamics: A Scalable Model with the Accuracy of Quantum Mechanics. *Phys. Rev. Lett.* **120**, 143001 (2018).
69. Konovalov, K. A., Unarta, I. C., Cao, S., Goonetilleke, E. C. & Huang, X. Markov State Models to Study the Functional Dynamics of Proteins in the Wake of Machine Learning. *JACS Au* **1**, 1330–1341 (2021).
70. Li, C., Liu, Z., Goonetilleke, E. C. & Huang, X. Temperature-dependent kinetic pathways of heterogeneous ice nucleation competing between classical and non-classical nucleation. *Nat. Commun.* **12**, 4954 (2021).
71. Wu, Y., Cao, S., Qiu, Y. & Huang, X. Tutorial on how to build non-Markovian dynamic models from molecular dynamics simulations for studying protein conformational changes. *J. Chem. Phys.* **160**, 121501 (2024).
72. Jorgensen, W. L., Maxwell, D. S. & Tirado-Rives, J. Development and Testing of the OPLS All-Atom Force Field on Conformational Energetics and Properties of Organic Liquids. *J. Am. Chem. Soc.* **118**, 11225–11236 (1996).
73. Li, H. *et al.* Nucleation and Growth of Covalent Organic Frameworks from Solution: The Example of COF-5. *J. Am. Chem. Soc.* **139**, 16310–16318 (2017).
74. Li, H. & Brédas, J.-L. Large Out-of-Plane Deformations of Two-Dimensional Covalent Organic Framework (COF) Sheets. *J. Phys. Chem. Lett.* **9**, 4215–4220 (2018).
75. Li, H. & Brédas, J.-L. Nanoscrolls Formed from Two-Dimensional Covalent Organic Frameworks. *Chem. Mater.* **31**, 3265–3273 (2019).
76. Li, H. & Brédas, J.-L. Impact of Structural Defects on the Elastic Properties of Two-Dimensional Covalent Organic Frameworks (2D COFs) under Tensile Stress. *Chem. Mater.* **33**, 4529–4540 (2021).

77. Pelkowski, C. E. *et al.* Tuning Crystallinity and Stacking of Two-Dimensional Covalent Organic Frameworks through Side-Chain Interactions. *J. Am. Chem. Soc.* (2023) doi:10.1021/jacs.3c03868.
78. Addicoat, M. A., Vankova, N., Akter, I. F. & Heine, T. Extension of the Universal Force Field to Metal–Organic Frameworks. *J. Chem. Theory Comput.* **10**, 880–891 (2014).
79. Kupgan, G., Chen, X. K. & Brédas, J. L. Molecular packing of non-fullerene acceptors for organic solar cells: Distinctive local morphology in Y6 vs. ITIC derivatives. *Mater. Today Adv.* **11**, 100154 (2021).
80. Wang, T. & Brédas, J.-L. Organic Photovoltaics: Understanding the Preaggregation of Polymer Donors in Solution and Its Morphological Impact. *J. Am. Chem. Soc.* **143**, 1822–1835 (2021).
81. Essmann, U. *et al.* A smooth particle mesh Ewald method. *J. Chem. Phys.* **103**, 8577–8593 (1995).
82. Bussi, G., Donadio, D. & Parrinello, M. Canonical sampling through velocity rescaling. *J. Chem. Phys.* **126**, 014101 (2007).
83. Berendsen, H. J. C., Postma, J. P. M., van Gunsteren, W. F., DiNola, A. & Haak, J. R. Molecular dynamics with coupling to an external bath. *J. Chem. Phys.* **81**, 3684–3690 (1984).
84. Pronk, S. *et al.* GROMACS 4.5: a high-throughput and highly parallel open source molecular simulation toolkit. *Bioinformatics* **29**, 845–854 (2013).

Reviewer Response Letter

Reviewer/Editorial comments are in **black**.

Our responses are shown in **blue**.

Changes or additions to the manuscript are shown in **green**.

Reviewer #1

I thank the authors for their response and the revised manuscript. However, I find that my primary concerns were not adequately addressed, and the core issues remain unresolved.

Reply:

We sincerely thank the Reviewer for the thoughtful evaluation and constructive comments, which have been instrumental in improving our manuscript. Upon reviewing their latest comments, we realized that certain points in our previous response were based on our misinterpretation of the Reviewer's original intent. We have carefully re-examined the relevant issues and provide our detailed responses below.

My specific concerns are outlined below:

1. HiDiscover is framed as a tool for uncovering structural arrangements relevant in understanding molecular mechanisms from MD data. However, in my opinion, there is absolutely no basis for HiDiscover to learn mechanistic insights for any dynamical processes, simply because it does not consider the dynamics of the data (acknowledged by the authors in their response: "Since our focus is on identifying atomic and molecular arrangements (static features),...". I raised this issue in my first round of review, to which the authors provided autocorrelation calculations. Unfortunately, this does not address the concern at all. I give an example to clarify what I mean: Imagine, there are two metastable states: (I) free CO₂, and (ii) adsorped CO₂. In the paper, the authors generated data from these two states and trained an ML model to comment on which features are relevant for the adsorption process (the training data included trajectories from more than two initial arrangements, but the idea behind this example stays true). However, to understand such mechanisms we need to look at the intermediate, transition state in-between these two states, where the relevant features can be completely different compared to which features are relevant in the two metastable states respectively. Thus, mechanistic insights, by definition, rely on the dynamic process connecting these states, and relevant features may arise only transiently during the transition. HiDiscover compares equilibrated configurations and draws conclusions that, in my view, do not meet the standards for mechanistic interpretation. This gets complicated since we typically don't know what/where is the transition state and its also short-lived which will likely get washed away and the system will collapse into a single metastable state when the authors do their equilibration+MD steps in HiDiscover.

Reply:

We thank the Reviewer for their detailed explanation, which prompted us to re-evaluate our previous response to Comment 1. Upon further reflection, we now realize that we did not fully grasp the essence of the Reviewer's concern. While our initial response focused on the influence of temporal correlations on the predictive accuracy of HiDiscover, we now understand that the Reviewer's original concern may have been that (1.1) uncovering material mechanisms using a new method requires that this method directly capture dynamic features, given that mechanisms are often regarded as inherently dynamic in nature. In this new comment, the Reviewer has further pointed out that (1.2) the intermediate states crucial to understanding the dynamics of these mechanisms may not be adequately represented in our MD simulations of equilibrium systems. We greatly appreciate this clarification, which has allowed us to refine and expand our response accordingly.

Since the term "state" was used extensively in the Reviewer's comment but can carry multiple interpretations, we would like to clarify its use here to avoid potential misunderstandings. In the

following discussion, we specifically use the term “microscopic state” to refer to distinct atomic and molecular configurations observed in the simulated systems. It is important to note that a system may consist of quasi-static (meta)stable microscopic states, or it may exist in dynamic equilibrium, continuously transitioning between different microscopic states. Moreover, a simulation may capture only a snapshot (or a short segment) of a longer dynamic process. Depending on the research objective, one may be interested in the dynamic transitions between microscopic states (which are sometimes accessible within the timescale of MD simulations), the longer-timescale processes that govern system evolution (which are often beyond MD’s reach), or both. We also note that equilibrium simulations are generally much more practical and widely adopted than the *non-equilibrium simulations*, whether involving transitions between microscopic states or longer-timescale dynamics.

Regarding point (1.1), this view seems to originate in a more specific concept of mechanisms than what is commonly adopted in recent studies of advanced materials, particularly those relevant to energy and environmental applications (which are the primary focus of HiDiscover). We fully agree that many material mechanisms are inherently dynamic. We also note that in the study of many complex, multi-component materials, mechanisms with dynamic characteristics are often not readily accessible from simulations due to various practical limitations. In practice, researchers frequently infer such mechanisms from static features of partially sampled structures. Moreover, many widely accepted mechanisms in the materials science community are, in fact, primarily static interpretations. The HiDiscover protocol can support mechanistic studies under these conditions.

Concerning point (1.2), some of the terminology used in the Reviewer’s comment appears to be different from conventions typically employed in recent modeling of advanced materials. To ensure a more inclusive interpretation and avoid potential misunderstandings, we propose two possible sources for the concern raised: (1.2.1) the newly introduced reference systems in HiDiscover may not exhibit intermediate microscopic states; (1.2.2) simulations of equilibrium systems represent only discrete “slices” along a complete dynamic trajectory, potentially missing certain features that occur between these quasi-static conditions.

We acknowledge that both points (1.1) and (1.2) reflect differing perspectives on how mechanisms are interpreted and investigated in contemporary research on complex, multi-component materials. In the following, we provide a detailed discussion of our viewpoint on *the concept of mechanisms in this context, the typical approaches by which mechanistic insights are obtained, and the ways in which the HiDiscover protocol can contribute to such studies*. We also address relevant considerations specific to simulations of complex materials, as these appear to be a potential source of misunderstanding.

(i) Due to challenges such as limited knowledge on (meta)stable microscopic states for constructing appropriate initial configurations, or technical difficulties associated with performing non-equilibrium MD simulations, many dynamic transitions between microscopic states in complex materials are instead studied by simulating the system under dynamic equilibrium, where such transitions occur naturally. If these transitions can be adequately modeled, are well-sampled in the MD trajectories, and manual examination of the microscopic structures is feasible, the changes between microscopic states may be observed, interpreted, and reported as mechanisms in a conventional MD study.

If the HiDiscover protocol is used, it is possible that we can only design new reference systems corresponding to the initial and final microscopic states (this might be related to the second point raised by the Reviewer, specifically point (1.2.1)). However, as the target system (which typically includes transitions among initial, intermediate, and final microscopic states) is also used as a reference system, the intermediate microscopic states can be identified and differentiated from the initial and final microscopic states. When the learned features are plotted along the simulation

trajectory, patterns indicative of dynamic transitions may emerge, provided that the interval between MD output frames is appropriate and the patterns are sufficiently discernible to the researcher. As such, while HiDiscover does not directly predict dynamics, it can aid in *identifying molecular features associated with dynamic behavior*, thereby reducing the burden of manual analysis.

(ii) It is also important to recognize that in many studies of advanced materials, mechanisms involving dynamic processes are not established directly from the simulation of those dynamics. Instead, researchers often deduce plausible mechanisms by analyzing the characteristics of quasi-static systems obtained from theoretical calculations, experimental observations, or a combination of both. There are many practical reasons for adopting this approach. A common one is that the dynamic processes of interest often occur on *timescales far exceeding the capabilities of molecular dynamics simulations*. In such cases, capturing the complete dynamics within a single simulation is infeasible, and the system often remains in a metastable condition. This may be the point the Reviewer was referring to in the final sentence of the comment, specifically point (1.2.2). This represents a general and well-recognized challenge in materials modeling—one that the HiDiscover protocol is not designed to overcome.

As a result, mechanisms involving dynamic processes are frequently inferred from the analysis of static features. A common practice in these cases is to simulate a series of equilibrium systems (sometimes complemented by experimental measurements to maximize the information obtained) and use the resulting data to propose a plausible mechanism along a hypothesized dynamic pathway. For example, to investigate the formation of the electric double layer (which is crucial for further understanding the formation of the solid-electrolyte interphase), electrolyte conformations near the anode can be independently simulated at various applied voltages from the charging process. The dynamic formation mechanisms of the electric double layer are then inferred from the static structural features extracted from these “snapshots” (*e.g.*, Nat. Nanotechnol., 2020, 15, 224). This mode of reasoning (inferring mechanisms from static or partially dynamic data) is widely adopted and generally well-accepted in the study of complex, multi-component materials. A significant bottleneck in these studies is the difficulty of identifying detailed atomic and molecular features, precisely the challenge that HiDiscover seeks to address.

(iii) Finally, we would like to emphasize that many well-established mechanisms in materials science are primarily static interpretations. In the study of advanced materials, the term “mechanism” is often used broadly to describe *the microscopic origins of material behaviors or properties*, and does not necessarily imply an emphasis on dynamic processes. Specifically, such mechanisms may refer to how structural characteristics or compositional factors influence material function, to correlations between structure and performance, without requiring an explicit description of the underlying dynamics. For example, in organic photovoltaics, the mechanism by which additives enhance performance has been attributed to improved π - π stacking in the active layer—a predominantly structural explanation (*e.g.*, Adv. Mater., 2024, 36, 2406623). Similarly, the mechanism underlying the improved electrochemical performance of a reconstructed surface has been attributed to its bulk-like morphology (a static interpretation), which is expected to facilitate uniform Li plating and stripping (*e.g.*, Energy Environ. Sci., 2024, 17, 260). (We would like to note that these “mechanisms”, while having a static focus, do not imply the absence of dynamic processes. In many cases, understanding static features is sufficient for a wide range of material design purposes. Moreover, investigating detailed dynamic processes may not always be preferred, either due to the *limited practical benefit or the lack of suitable tools* to effectively probe them.) In these contexts, identifying structural features in multi-component systems remains a critical challenge. By uncovering and quantifying such features, HiDiscover provides a route to elucidate static mechanisms that are useful for materials design.

With the above clarification, we would like to note that we did not attempt to use the HiDiscover protocol to directly learn the dynamics of CO₂ motion from the MD simulations. As stated on page 20 of the main text, our analysis was intentionally focused on static structural features: “Here, we demonstrate that using the HiDiscover protocol allows us to reveal in detail and quantitatively the arrangement features of CO₂ in the representative MOF-5.” We did briefly discuss the possible dynamic behavior of CO₂ in MOF-5, which was based on general reasoning informed by established knowledge in materials science. We have carefully reviewed the corresponding wording to ensure that it does not overstate our conclusions or imply otherwise.

For clarity, we have revised the sentence on page 3 to read: “At this point in time, mechanistic insight (whether focused on static or dynamic features) is largely derived from molecular simulations,⁷ which determine the evolution of atomic coordinates that are denoted as trajectories.” Additionally, we have included the following clarification on page 3: “The study of these advanced materials is already complicated by slow dynamics and/or complex factors that often hinder full characterization via a single simulation. Consequently, mechanistic insights are frequently inferred from data available to the researcher.” We have also updated the wording on page 9 as follows: “To promote its application as a supportive tool in the investigation of microscopic mechanisms in materials, we have developed a research protocol named HiDiscover (Figure 1c).”

We hope this clarification satisfactorily addresses the Reviewer’s concerns and clearly conveys the rationale behind the design and applicability of the HiDiscover protocol in supporting mechanistic studies of complex, multi-component materials.

2. I strongly disagree with the authors’ claim that “In fact, the HiDiscover framework does not require additional prior information compared to a conventional MD study of multi-component materials...” In a conventional MD simulation, the dynamics and relevant configurations naturally emerge from first principles based on the equations of motion and an initial arrangement. In contrast, HiDiscover requires the user to supply additional reference arrangements, effectively directing the protocol toward specific features in the trajectory. This imposes a high level of system-specific prior knowledge and in my opinion reduces the method’s general utility. I note that the other reviewer also raised similar concerns.

Reply:

The quoted sentence was from our previous response letter addressing the original Comment 2. After carefully reviewing the new comment, we now have a clearer understanding of the Reviewer’s initial concern. In retrospect, we recognize that our previous statement may have appeared too strong. We agree with the Reviewer that, strictly speaking, the HiDiscover protocol does introduce an additional layer of prior information, as it requires the explicit construction of reference systems. A more appropriate statement should be that, in the context of mechanistic studies of multi-component materials, the *effort* involved in gathering this additional prior information (used to construct new reference systems that have simpler configurations) is usually minor when considering the amount of prior information already gathered for the target system in a conventional MD study.

Before elaborating on the effort involved in obtaining additional prior information for the new reference systems in HiDiscover, we would first like to clarify the extent to which prior information is typically required to construct the initial configuration of the target system in conventional MD simulations, as this might be a source of misunderstanding.

(i) For very simple systems such as a small, rigid molecule with limited degrees of freedom (*e.g.*, pentacene, a classical organic semiconductor with five fused benzene rings), setting up an initial atomic configuration requires only basic knowledge of the molecular structure and typical bond lengths and angles, which is often obtainable from databases or software tools such as GaussView.

(ii) However, for complex materials, the situation becomes considerably more challenging. These systems often exhibit spatial heterogeneity and/or sluggish dynamics. Under such circumstances, constructing a physically reasonable initial configuration becomes infeasible without acquiring prior information in a more rigorous way. Consider the example of a pentacene crystal, which represents a slightly more complex case compared to its isolated molecular form. Experimental crystallographic data (*e.g.*, from XRD) are essential to guide the construction of the initial structure. This need arises because current MD simulations are typically limited to nanosecond timescales, which are very far from sufficient for the system to evolve from an arbitrary configuration into its most stable crystalline structure. Such limitations are a common challenge in the modeling of more complex multi-component materials. Due to the slow dynamics often involved (*e.g.*, component mixing, interface formation, or phase transitions), it is usually infeasible to obtain realistic structures solely through MD-based temporal evolution. As a result, *these structural features are often incorporated into the initial configuration based on prior knowledge.*

With this clarification, we would like to emphasize that *gathering as much relevant prior knowledge as possible* is a well-established practice in conventional MD modeling of advanced multi-component materials. *The characteristics of the individual components are usually an essential part of this prior knowledge*, as they inform the construction of the initial configuration of the multi-component material. In this context, when building additional reference models (such as those of the components already present in the system, as we have suggested), *the effort required to obtain further prior information is often minimal or marginal.*

It appears that some of the terminology in the comments may differ from conventions commonly adopted in recent modeling of advanced materials, particularly those relevant to energy and environmental applications that HiDiscover is mainly designed for. To ensure that we have not overlooked any key points and to avoid potential misunderstandings, we would like to offer the following additional discussion and clarification:

(iii) In the HiDiscover protocol, it is not necessary to design new reference systems to capture all relevant structures present in the target material. As we explained already in our response to Comment 1, the target system itself is included in the list of reference systems, ensuring that all relevant structural conformations will be learned. The atomic and molecular arrangements originating from the introduced reference systems can be related to features in simpler materials, while those that only exist in the target system will be identified as new features in the multi-component material. We would like to note that *whether the identified features are “known” or “new” provides valuable information* in the context of contemporary studies on complex, multi-component materials.

(iv) We would like to emphasize that the introduction of reference systems does not guide or constrain the actual simulation of the target system. The simulation proceeds independently, governed solely by the equations of motion from the initial configuration.

(v) The level of structural differentiation provided by the reference systems influences the granularity of information that can be extracted using HiDiscover. As a starting point, researchers can typically use the constituent components of the multi-component material as reference systems; this generally requires minimal effort in obtaining additional prior knowledge, as previously discussed. If finer structural differentiation is desired, additional reference systems can be constructed, which may require greater effort. This strategy is analogous to that employed in conventional MD studies, where researchers design additional reference systems to enable more comprehensive comparisons and extract more detailed insights, which likewise require increased effort. Nevertheless, regardless of the degree of detail, the insights gained from HiDiscover are expected to offer added value for understanding the mechanisms of multi-component materials, beyond what conventional analyses typically provide.

We hope the above explanations have adequately addressed the Reviewer's concerns regarding the prior information associated with the HiDiscover protocol.

We have confirmed that the sentence previously cited by the Reviewer appears only in our previous response letter and not in the manuscript itself. To avoid any potential misunderstanding, we have revised the wording on page 10 to read: "We note that, when a HiDiscover protocol is being considered instead of a conventional MD study, a good background knowledge of the studied material (e.g., structural features of its components) is already or will be obtained and a well-defined objective for the researcher to elucidate specific material mechanisms has been or will be established. As such, the ultimate design of the reference systems is often feasible, as we demonstrate in several distinct examples." We also added the following sentence on page 28: "In practice, applying the HiDiscover protocol does not seem to substantially increase the effort required to obtain prior knowledge, as demonstrated in three distinct examples."

3. The authors state that they are exploring the feasibility of "substituting conventional MD study" with the HiDiscover framework. While such a capability would indeed be a significant advance, unfortunately the current manuscript offers no compelling evidence or results to support this claim.

Reply:

We appreciate the Reviewer's careful reading and apologize for any confusion we might have caused. The statement in question was included only in our previous response letter, specifically in reply to the original Comment 2. The Reviewer's further comments made us recognize that our claim was not accurate, as our current work does not encompass a comprehensive demonstration across all possible scenarios. Thus, it is indeed premature at this stage to suggest that HiDiscover could fully substitute conventional MD studies.

To clarify, we have thoroughly re-examined the manuscript and confirmed that this particular aspect does not appear in the main text. As indicated above, it was only present in the previous response letter. Moreover, we have carefully reviewed the manuscript and made sure that our wording would not lead to similar misunderstandings.

Reviewer #2

I appreciate the authors' thorough revisions to the manuscript in response to the referees' comments. The authors have successfully addressed all the concerns raised, leading to significant improvements in the clarity and quality of the manuscript. The value of the HiDiscover protocol, particularly its ability to facilitate detailed differentiation and efficient extraction of ionic and molecular arrangements that are difficult to identify using conventional molecular simulations, is now clearly demonstrated. This conclusion is well supported by the presented results and discussions. Accordingly, I recommend the manuscript for publication in Nature Communications.

Reply:

We sincerely thank the Reviewer for their positive feedback and insightful suggestions, which have been highly valuable in helping us improve the quality of our manuscript.

Reviewer Response Letter

Reviewer comments are in **black**.
Our responses are shown in **blue**.

Reviewer #1

I thank the authors for their response to my comments, which I appreciate. My final thoughts on the work are as follows:

From a theoretical/methodological standpoint, this work would have represented a significant advance if dynamic interpretations were generated by treating MD data as a time-series as I mentioned in the previous rounds. However, the authors clarified that they are aiming for static interpretations of molecular arrangements. Unfortunately, the field has become saturated with machine-learning driven static interpretation schemes, and in my view, this work for multi-component materials does not substantially advance the field to warrant broad impact, and the results are not fully convincing.

The proposed protocol involves many engineering steps that, as I noted in my initial reviews, are difficult to translate into actionable strategies for a user.

Reply:

We fully agree with the Reviewer that machine learning approaches have been widely applied in studies involving molecular simulations. Our central argument in this work is that, in the context of mechanistic investigations of a wide range of multi-component materials frequently encountered in energy and environmental research, there remains a lack of methodologies that can assist researchers in systematically extracting features of atomic and molecular arrangements, particularly features that are easily interpretable. Consequently, researchers often resort to manual inspection of simulation trajectories, typically in a non-exhaustive manner, and, in certain cases, may be unable to obtain meaningful insights due to the inherent complexity of the atomic structures. In this work, we introduce a hierarchical incremental learning protocol, HiDiscover, that facilitates the extraction of microscopic structural features in a manner that is both accessible to researchers and conducive to a comprehensive analysis of simulation data. This capability makes the protocol particularly valuable for mechanistic studies of complex materials, as demonstrated through three distinct examples. We therefore anticipate that HiDiscover will advance the study of such materials and exert a broad influence on contemporary materials modeling strategies for complex systems.

The HiDiscover protocol comprises several steps, some of which are newly introduced as a result of integrating machine learning techniques (*e.g.*, dataset construction, model training). Other steps are conceptually present, albeit implicitly, in many conventional MD studies (*e.g.*, gathering preliminary knowledge, defining tasks and reference systems). We understand that the Reviewer may be concerned with the feasibility of the latter steps. In HiDiscover, these steps are made explicit, in contrast to their often implicit treatment in traditional MD workflows. This explicit formulation does not hinder the protocol's practicality, as explained earlier and discussed in the manuscript. In conventional MD studies, reference systems are already frequently employed. When applying HiDiscover to multi-component materials, such reference systems can simply correspond to the individual components already present in the material. Similarly, the tasks are similar to the "points of interest" that researchers focus on during simulation analysis. We believe that the examples provided in the manuscript serve as a practical guide for implementing the HiDiscover protocol in future research.